# The effect of lithology on the relationship between denudation rate and chemical weathering pathways. Evidence from the eastern Tibetan Plateau.

Aaron Bufe[1], Kristen L. Cook[1], Albert Galy[2], Hella Wittmann[1], Niels Hovius[1,3]

[1]German Research Center for Geosciences, Potsdam 14473, Germany
[2]Centre de Recherches Pétrographiques et Géochimiques, CNRS, Université de 13 Lorraine, 54500 Nancy, France.
[3]Institute of Geosciences, Potsdam University, Potsdam 14476, Germany

*Correspondence to*: Aaron Bufe (aaronbufe@gmail.com)

**Abstract.** The denudation of rocks in mountain belts exposes a range of fresh minerals to the surface of the Earth that are chemically weathered by acidic and oxygenated fluids. The impact of the resulting coupling between denudation and weathering rates fundamentally depends on the types of minerals that are weathering. Whereas silicate weathering sequesters $CO_2$, the combination of sulfide oxidation and carbonate dissolution emits $CO_2$ to the atmosphere. Here, we combine the concentrations of dissolved major elements in stream waters with $^{10}Be$ basin-wide denudation rates from 35 small catchments in eastern Tibet to elucidate the importance of lithology in modulating the relationships between denudation rate, chemical weathering pathways, and $CO_2$ consumption or release. Our catchments span three orders of magnitude in denudation rate in low-grade flysch, high grade metapelites, and granitoid rocks. For each stream, we estimate the concentrations of solutes sourced from silicate weathering, carbonate dissolution, and sulfide oxidation using a mixing model. We find that for all lithologies, cation concentrations from silicate weathering are largely independent of denudation rate, but solute concentrations from carbonates and, where present, sulfides increase with increasing denudation rate. With increasing denudation rates, weathering may, therefore, shift from consuming to releasing $CO_2$ in both (meta)sedimentary and granitoid lithologies. For a given denudation rate, we report 2 – 10-times higher dissolved solid concentrations and inferred weathering fluxes in catchments underlain by (meta)sedimentary rock compared to catchments containing granitoid lithologies, even though climatic and topographic parameters do not vary systematically between these catchments. Thus, varying proportions of exposed (meta)sedimentary and igneous rocks during orogenesis could lead to changes in the sequestration and release of $CO_2$ that are independent of denudation rate.

## 1 Introduction

The relationship between chemical weathering and $CO_2$ drawdown modulates the global carbon cycle and Earth's climate (Berner et al., 1983; Walker et al., 1981). Uplift and denudation of rock control the supply of unweathered minerals to the surface of the Earth and impact the residence time of these minerals in the weathering zone (e.g. Gabet and Mudd, 2009; Hilley

et al., 2010; Riebe et al., 2001; West et al., 2005). Thus, increases in denudation rates can boost chemical weathering and ultimately affect Earth's climate (Caves Rugenstein et al., 2019; Hilton and West, 2020; Kump and Arthur, 1997; Raymo and Ruddiman, 1992). The impact of chemical weathering on the attendant emission or drawdown of $CO_2$ depends on the relative importance of different weathering pathways (Berner et al., 1983; Bufe et al., 2021; Calmels et al., 2007; Hartmann et al.,

2009; Hilton and West, 2020; Torres et al., 2016; Torres et al., 2014). Weathering of silicate minerals by carbonic acid consumes $CO_2$ from the atmosphere and drives the sequestration of this carbon over timescales longer than marine carbonate compensation (Walker et al., 1981). In turn, sulfuric acid produced by the oxidation of pyrite can drive rapid dissolution of carbonate minerals that releases $CO_2$ into the atmosphere over timescales that exceed the timescale of carbonate compensation in the Ocean (Calmels et al., 2007; Das et al., 2012; Spence and Telmer, 2005; Torres et al., 2014), although the impact of that

inorganic carbon source and its link to the organic carbon cycle remain debated (Maffre et al., 2021; Torres et al., 2014). Weathering of silicate minerals by sulfuric acid and of carbonate minerals by carbonic acid are carbon neutral (Calmels et al., 2007; Spence and Telmer, 2005). Importantly, carbonates and sulfides weather several orders of magnitude faster than silicate minerals (Berner, 1978; Morse and Arvidson, 2002; Williamson and Rimstidt, 1994), such that sulfuric acid-driven carbonate weathering dominates the total solute flux even in lithologies with small fractions of pyrite or carbonate (Anderson et al., 2000;

Bufe et al., 2021; Calmels et al., 2007; Das et al., 2012; Emberson et al., 2016b; Jacobson and Blum, 2003; Kemeny et al., 2021; Relph et al., 2021; Spence and Telmer, 2005; Torres et al., 2016).

The dependence of chemical weathering on mineralogy implies a lithologic control on the balance of $CO_2$ drawdown and $CO_2$ release by weathering reactions. Fresh and fine-grained mafic rocks, such as basalts, exhibit some of the fastest silicate weathering rates on the planet and contribute disproportionally to global silicate weathering fluxes (Dessert et al., 2003;

Gaillardet et al., 1999; Ibarra et al., 2016; Li et al., 2016). Felsic igneous rocks typically weather more slowly, but silicate weathering in these rocks is also thought to be sensitive to denudation rates (Riebe et al., 2004; West et al., 2005). In contrast, weathering of siliciclastic (meta)sedimentary rocks is often dominated by the dissolution of minor carbonate and sulfides (Anderson et al., 2000; Blattmann et al., 2019; Bufe et al., 2021; Calmels et al., 2007; Das et al., 2012; Emberson et al., 2016b; Jacobson and Blum, 2003; Kemeny et al., 2021; Relph et al., 2021; Spence and Telmer, 2005; Torres et al., 2016), and rapid

denudation and exposure of these rocks can drive $CO_2$ release from weathering (Bufe et al., 2021; Calmels et al., 2007; Märki et al., 2021; Spence and Telmer, 2005; Torres et al., 2016; Torres et al., 2014). Carbonate dissolution also dominates weathering of carbonate-rich sediments (Erlanger et al., 2021; Gaillardet et al., 2018; Gaillardet et al., 1999), but the role of sulfide oxidation and, therefore, the impact of carbonate weathering on the long-term $CO_2$-cycle can be limited in these rocks (Erlanger et al., 2021; Gaillardet et al., 2018; Gaillardet et al., 1999).

Even though lithology exerts a fundamental control on the link between weathering and Earth's carbon cycle, few studies have directly compared and quantified the impact of lithologic variation on the link between physical denudation and chemical weathering and on the importance of different weathering pathways. Such knowledge is important in the context of mountain building, as ongoing exhumation progressively exposes successions of different lithologies to chemical weathering (Hilton

and West, 2020). Landscape-scale weathering fluxes are typically estimated by measuring the solute concentration and runoff

in rivers that integrate water fluxes across the upstream catchment (Drever and Clow, 2018; Drever and Zobrist, 1992; Gaillardet et al., 1999; White and Blum, 1995). In many settings, denudation gradients coincide with gradients in precipitation, temperature, tectonics, biomass, and/or substrate properties such as fracture density or metamorphic grade (Dixon et al., 2016); gradients that affect weathering kinetics independently of denudation rates (Drever and Zobrist, 1992; Gaillardet et al., 2018; Godsey et al., 2019; Guo et al., 2019; Ibarra et al., 2016; Li et al., 2016; Oeser and von Blanckenburg, 2020; Uhlig and von

Blanckenburg, 2019; White and Blum, 1995). These co-variations can complicate the interpretation of changes in weathering. Here, we investigate the impact of lithologic variations on chemical weathering across a nearly three-order of magnitude wide denudation rate gradient along the eastern margin of the Tibetan Plateau that includes three well-separated groups of lithology (Fig. 1) and relatively small changes in precipitation, elevation, and temperature (Fig. 2, Table S1).

## 2 Geologic setting

On the eastern Tibetan Plateau west of the Sichuan Basin lies the Songpan-Ganze (or Songpan-Garzê) fold belt, which formed during the late Triassic to early Jurassic closure of part of the Paleotethys ocean (Burchfiel et al., 1995; Chen et al., 2007; Weller et al., 2013). The Songpan-Ganze terrane is characterized by a 5 – 15 km thick sequence of tightly folded Triassic flysch that overlies Paleozoic metasedimentary rocks and Precambrian basement (Roger et al., 2010; Weller et al., 2013) (Fig. 1). These units have been intruded by granitoids during both the Mesozoic and Cenozoic. Reactivation of the region during

the Cenozoic Indo-Eurasian collision formed zones of rapid uplift and denudation within and at the eastern edge of the slowly eroding Tibetan plateau. Here, we focus on the southeastern part of the Songpan-Ganze fold belt in the region around the Danba Structural culmination and the zone of rapid uplift centred around Gongga Shan (Fig. 1). In this area, in-situ [10]Be-derived basin-wide denudation rates vary more than 200-fold over a distance of 100-200 km, from 0.018 mm/yr on the plateau margin (Fig. 1) to 7 mm/yr around Mt Gongga and the Danba structural culmination (Cook et al., 2018; Ouimet et al., 2009)

(Fig. 1). This variation of denudation rates is reflected in the topography with variations of basin averaged relief between 340 m – 5260 m and basin-averaged slopes of 10° – 37.9°. The region features three principal lithologic groups. Most of the area exposes the Triassic Songpan-Ganze flysch, a sequence of deep sea turbidites that is unmetamorphosed or weakly metamorphosed over large areas but can reach greenschist facies close to Gongga Shan and the Danba Structural culmination (Burchfiel et al., 1995; Chen et al., 2007). The underlying Paleozoic sequence is exposed in the Danba structural culmination

and in the Longmen Shan and consists of a passive margin sequence, which has undergone metamorphism up to amphibolite facies (Huang et al., 2003; Weller et al., 2013). Finally, granitic rocks of a range of ages from Precambrian to Cenozoic are exposed throughout the area (Roger et al., 2010; Roger et al., 2004; Searle et al., 2016) (Fig. 1, Table S1). Rocks within each one of these three broad lithologic groups have varying compositions, but the difference in chemical compositions and mineralogy between these three lithologic groups is greater than the variability within each one of the groups (Chen et al.,

2007; Jiang et al., 2018; Weller et al., 2013) (Table S3, Fig. 3). Moreover, minor carbonate and sulfide phases occur in varying proportions in these dominantly silicate lithologies (Chen et al., 2007; Jiang et al., 2018; Weller et al., 2013).

The region is characterized by a cold and humid climate with a pronounced wet monsoon season during the summer months and drier winter months. Mean annual precipitation rates derived by the Tropical Rainfall Measuring Mission (TRMM) vary ~4-5-fold between 300 – 1340 mm/yr (Bookhagen and Burbank, 2010) (Fig. 2b). This range of precipitation values is very similar to the range of runoff values previously estimated for small catchments around the Gongga Shan (Table 1 in Jiang et al. (2018)). Our sampled catchments feed the Dadu, Min, and the Yalongjiang rivers, tributaries to the Yangtze river. They are all headwater catchments that fall within a relatively narrow elevation window with catchment averaged elevations of 3500-4800 m with three exceptions at lower mean elevations of 2600 – 2800 m (Fig. 2a, Table S1). The treeline lies between ~3500 – 4100m. Thus, the majority of catchments spans across the treeline and includes forested areas, grasslands and rocky slopes. Neither the elevation, nor the annual precipitation in the studied catchments or the temperature of sampled stream waters correlate strongly with the denudation rate across the area (Fig. 2)

## 3 Methods

### 3.1 Sample collection and analysis

We collected water samples from 35 small catchments around the region of Mt Gongga and the Danba structural culmination (Fig. 1, Table S1). Water samples were collected in mid-May 2018, prior to the rainy season, over a period of 5 days when all catchments were subject to similar hydrological conditions. Most of the sampled catchments drain a single lithologic group: Thirteen catchments drain the Triassic flysch that is dominantly composed of unmetamorphosed or weakly metamorphosed mixed carbonate-siliciclastic sediments. Only two of the flysch catchments include rocks that have been metamorphosed to garnet-bearing greenschists (S18-35 & S18-36, Fig. 3c). Eleven catchments drain granitic rocks, and six catchments drain high-grade Paleozoic and Precambrian metasedimentary rocks, whereas five catchments drain a mix of granitic rocks and Triassic flysch (Fig. 3c, Table S1). Samples were taken as close to sites sampled for cosmogenic nuclide-derived denudation rates (Cook et al., 2018) as possible, and upstream of potential major anthropogenic influences. Water samples were filtered in situ with single-use 0.2 μm filters and collected in HDPE bottles. Aliquots for cation analysis were acidified on-site to a pH of ~2 with 3M ultrapure $HNO_3$. Temperature, pH, and conductivity were measured in the field using a WTW Multi 3430 multimeter.

Anion and cation analyses were conducted at the GFZ Potsdam on a Dionex ICS-1100 Ion Chromatograph and a Varian 720 ICP-OES, respectively, following the procedure described in Bufe et al. (2021). The concentration of bicarbonate was estimated by charge balance. Such estimation is reasonable in active mountain settings where organic acids are scarce and estimates of bicarbonate from charge balance are within uncertainty of alkalinity values from titrations (Galy and France-Lanord, 1999). Analytical uncertainties for cation analyses were derived from the largest deviation of the calibration standards

from the calibration line. For anions, uncertainty estimates were derived from the standard deviation of three repeat measurements (Table S2).

## 3.2 Unmixing of solute sources

We consider four major sources of cations to rivers in the study area: Silicate weathering, carbonate weathering, cyclic contributions (assumed to be dominated by wet precipitation), and hotsprings contributions (Table 1). We assume that evaporites are a minor component of the dissolved solids because: (i) Existing petrologic and geochemical studies do not report evaporite deposits in the upper Triassic flysch or in the Paleozoic metamorphic rocks around the Danba Structural culmination (Chen et al., 2007; Jiang et al., 2018), (ii) except for one sample, chloride concentrations range between 4-20 µmol/L and are thus well within the range of concentrations expected for atmospheric chloride input (Gaillardet et al., 1999), and (iii) the ratios of $\left[\frac{Ca^{2+}}{Na^+}\right]$ concentrations are higher than expected for typical evaporite deposits (Chen et al., 2007; Gaillardet et al., 1999; Jiang et al., 2018). Moreover, we assume that secondary precipitation of carbonate minerals is negligible in the area. This assumption is based on the observation that all samples are either undersaturated with respect to calcium carbonate or are within uncertainty of saturation, except for two Flysch basins that suggest slightly super-saturated conditions (Fig. A1).

We used a 4-endmember inverse model to unmix these contributions based on the relative concentration of the three major soluble cations ($Na^+$, $Ca^{2+}$, and $Mg^{2+}$) and chloride ($Cl^-$) (Bufe et al., 2021; Gaillardet et al., 1999; Kemeny and Torres, 2021; Moon et al., 2014; Torres et al., 2016). The mixing model requires that each one of the endmembers is unique and, therefore, occupies a range of compositions that is distinct from the composition of other endmembers in the considered space (Fig. 3a-b). In the model, we minimized the following set of equations using the constrained linear least squares solver lsqlin in MATLAB:

$$\left[\frac{X}{Ca^{2+}}\right]_{spl} = \alpha_{Ca,sil}\left[\frac{X}{Ca^{2+}}\right]_{sil} + \alpha_{Ca,carb}\left[\frac{X}{Ca^{2+}}\right]_{carb} + \alpha_{Ca,cy}\left[\frac{X}{Ca^{2+}}\right]_{cy} + \alpha_{Ca,hs}\left[\frac{X}{Ca^{2+}}\right]_{hs}, \tag{1-3}$$

With $X = Na^+$, $X = Mg^{2+}$, and $X = Cl^-$ (three equations) under the conditions that:

$$\alpha_{Ca,sil} + \alpha_{Ca,carb} + \alpha_{Ca,cy} + \alpha_{Ca,hs} = 1 \tag{4}$$

and

$$0 \leq \alpha_{Ca,sil} \leq 1; \ 0 \leq \alpha_{Ca,carb} \leq 1; \ 0 \leq \alpha_{Ca,cy} \leq 1; \ 0 \leq \alpha_{Ca,hs} \leq 1 \tag{5}$$

Here, $\left[\frac{X}{Ca^{2+}}\right]$ is the ratio of concentrations of ion $X$ and calcium, and $\alpha_{Ca,Y}$ is the fraction of calcium sourced from endmember $Y$. The subscript $spl$ denotes the sampled ratio in the river water. The endmember subscripts $sil$, $carb$, $cy$, and $hs$ denote contributions from silicate, carbonate, cyclic (precipitation), and hotspring sources respectively.

Equations 1-5 were then solved separately for each of the 35 samples based on the input endmember compositions ($\left[\frac{X}{Ca^{2+}}\right]_{sil,carb,cy,hs}$, see Table 1) and the measured ratios ($\left[\frac{X}{Ca^{2+}}\right]_{spl}$, see Table S2). Note that one out of three different silicate endmembers was chosen for each sample, depending on the lithology in the sampled catchments (Table 1, Table S1). Where catchments contain a mix of different lithologies (for example S18-22 and S18-23, see Table S2), the input silicate endmember was determined by the average between the silicate endmembers from the lithologies weighted by the areal coverage of each lithology class in the catchment (see Table S1).

In order to represent the uncertainty of the endmember compositions in the inversion result, we implemented a Monte Carlo approach and ran 100,000 iterations of the inversion for each sample. For each iteration, we randomly picked the endmember from a normal distribution with mean and standard deviation defined by the endmember estimate and its uncertainty (Table 1). For the hotspring endmember, we instead picked from a uniform distribution defined by a minimum and maximum estimate (see rational below, Table 1). For each sample (35 samples) and each iteration (100,000 iterations) we used a reduced chi-squared statistic to estimate the goodness of fit of that sample and iteration. The goodness of fit, $\chi^2_{total}$, was based on (1) $\chi^2_{spl}$, the squared distances between elemental ratios predicted by that particular iteration $\left[\frac{X}{Ca^{2+}}\right]^{model}_{spl}$ and the sampled elemental ratios $\left[\frac{X}{Ca^{2+}}\right]_{spl}$ normalized by the uncertainty in the sampled ratio $\sigma_{\left[\frac{X}{Ca^{2+}}\right]_{spl}}$ (see Table S2 for data and uncertainties) and (2) $\chi^2_{endmember}$, the squared distances between the endmember ratios picked from the normal distribution $\left[\frac{X}{Ca^{2+}}\right]^{model}_{sil,carb,cy,hs}$ and the average $\left[\frac{X}{Ca^{2+}}\right]_{sil,carb,cy,hs}$ endmember value normalized by the uncertainty of the endmember values $\sigma_{\left[\frac{X}{Ca^{2+}}\right]_{sil,carb,cy,hs}}$ (see Table 1 for endmember values and uncertainties) with:

$$\chi^2_{spl} = \left(\frac{\left[\frac{Na^+}{Ca^{2+}}\right]^{model}_{spl} - \left[\frac{Na^+}{Ca^{2+}}\right]_{spl}}{\sigma_{\left[\frac{Na^+}{Ca^{2+}}\right]_{spl}}}\right)^2 + \left(\frac{\left[\frac{Mg^{2+}}{Ca^{2+}}\right]^{model}_{spl} - \left[\frac{Mg^{2+}}{Ca^{2+}}\right]_{spl}}{\sigma_{\left[\frac{Mg^{2+}}{Ca^{2+}}\right]_{spl}}}\right)^2 + \left(\frac{\left[\frac{Cl^-}{Ca^{2+}}\right]^{model}_{spl} - \left[\frac{Cl^-}{Ca^{2+}}\right]_{spl}}{\sigma_{\left[\frac{Cl^-}{Ca^{2+}}\right]_{spl}}}\right)^2 \tag{6}$$

and

$$\chi^2_{endmember} = \left(\frac{\left[\frac{Na^+}{Ca^{2+}}\right]^{model}_{sil} - \left[\frac{Na^+}{Ca^{2+}}\right]_{sil}}{\sigma_{\left[\frac{Na^+}{Ca^{2+}}\right]_{sil}}}\right)^2 + \left(\frac{\left[\frac{Mg^{2+}}{Ca^{2+}}\right]^{model}_{sil} - \left[\frac{Mg^{2+}}{Ca^{2+}}\right]_{sil}}{\sigma_{\left[\frac{Mg^{2+}}{Ca^{2+}}\right]_{sil}}}\right)^2 + \left(\frac{\left[\frac{Mg^{2+}}{Ca^{2+}}\right]^{model}_{carb} - \left[\frac{Mg^{2+}}{Ca^{2+}}\right]_{carb}}{\sigma_{\left[\frac{Mg^{2+}}{Ca^{2+}}\right]_{carb}}}\right)^2 +$$

$$\left(\frac{\left[\frac{Na^+}{Ca^{2+}}\right]^{model}_{cy} - \left[\frac{Na^+}{Ca^{2+}}\right]_{cy}}{\sigma_{\left[\frac{Na^+}{Ca^{2+}}\right]_{cy}}}\right)^2 + \left(\frac{\left[\frac{Mg^{2+}}{Ca^{2+}}\right]^{model}_{cy} - \left[\frac{Mg^{2+}}{Ca^{2+}}\right]_{cy}}{\sigma_{\left[\frac{Mg^{2+}}{Ca^{2+}}\right]_{cy}}}\right)^2 + \left(\frac{\left[\frac{Cl^-}{Ca^{2+}}\right]^{model}_{cy} - \left[\frac{Cl^-}{Ca^{2+}}\right]_{cy}}{\sigma_{\left[\frac{Cl^-}{Ca^{2+}}\right]_{cy}}}\right)^2 \tag{7}$$

where

$$175 \quad \left[\frac{X}{Ca^{2+}}\right]^{model}_{spl} = \alpha^{model}_{Ca,sil} \left[\frac{X}{Ca^{2+}}\right]^{model}_{sil} + \alpha^{model}_{Ca,carb} \left[\frac{X}{Ca^{2+}}\right]^{model}_{carb} + \alpha^{model}_{Ca,cy} \left[\frac{X}{Ca^{2+}}\right]^{model}_{cy} + \alpha^{model}_{Ca,hs} \left[\frac{X}{Ca^{2+}}\right]^{model}_{hs} \qquad (8)$$

The superscript $model$ refers to quantities that apply to a particular iteration and a particular sample. Note that the endmember misfit $\chi^2_{endmember}$ ignores the contribution of the hotspring endmember because this endmember was picked from a uniform distribution. Similarly, $\left[\frac{Cl^-}{Ca^{2+}}\right]_{sil} = \left[\frac{Cl^-}{Ca^{2+}}\right]_{carb} = \left[\frac{Na^+}{Ca^{2+}}\right]_{carb} = 0$ (see Table 1), and these ratios are not considered in the endmember misfit. The two misfits were then summed and weighted by the number of unique squared distances (three distances for $\chi^2_{spl}$ – see equation 6 – and six distances for $\chi^2_{endmember}$ - see equation 7) to give equal weight to the summed sample and endmember misfits:

$$\chi^2_{total} = \frac{\chi^2_{spl}}{3} + \frac{\chi^2_{endmember}}{6} \qquad (9)$$

The choice of this weighting is arbitrary, and the absence of any weighting does not change the results substantially. For each sample, the best fit set of endmembers and corresponding fractions, $\alpha_{Ca}$, was then chosen from the iteration with the lowest total misfit, $\chi^2_{total}$. The uncertainty in these parameters was estimated from all Monte Carlo runs that fit the data within a threshold of $\chi^2_{total} \leq 1$ (on average 9% of runs). The high number of runs above the threshold of $\chi^2_{total} > 1$ is linked to our approach of picking groups of endmembers independently for each Monte Carlo iteration from the entire endmember space (see example in Fig. A2) without an optimization (e.g. Moon et al., 2014).

The most important parameters in the mixing model are the endmember estimates (Table 1). Both, silicate and carbonate endmembers are based on compiled bedrock compositions around the study area (Chen et al., 2007; Jiang et al., 2018; Weller et al., 2013) (Table S3) as well as the assumption that: $\left[\frac{Cl^-}{Ca^{2+}}\right]_{sil} = \left[\frac{Cl^-}{Ca^{2+}}\right]_{carb} = \left[\frac{Na^+}{Ca^{2+}}\right]_{carb} = 0$. Published bulk rock analyses of rocks in the region show a substantial variability. Nevertheless, samples from the three individual lithologic groups (granitoids, Paleozoic passive margin metapelites and Triassic flysch) form distinct trends in $Mg^{2+}/Ca^{2+}$ – $Na^+/Ca^{2+}$ space (Fig. 3a-b). The composition of the granitoids varies around the global silicate endmember estimated from small streams that drain pure silicate lithologies (Burke et al., 2018; Gaillardet et al., 1999). Compared to the granites, the Paleozoic metapelites are substantially enriched in magnesium and the flysch rocks are characterized by high calcium contents (Fig. 3a-b). Major carbonate phases are not reported in the Paleozoic metapelites (Weller et al., 2013) and in the granitoids (Jiang et al., 2018). Even if trace carbonate minerals can strongly affect solutes sourced from chemical weathering (Anderson et al., 2000; Blum et al., 1998; Emberson et al., 2016b; Jacobson and Blum, 2003), the bulk-rock geochemistry measurement should be affected only negligibly. Thus, we assume that the bulk rock compositions are representative of the silicate endmember from these lithologies, and we use the mean and standard deviation of the elemental ratios from individual bedrock samples as an estimate for the silicate endmember of that bedrock group (Table 1). In turn, the flysch samples contain 4% – 80% carbonate that occurs both as calcite minerals and as carbonate cement (Chen et al., 2007). Therefore, these samples represent a mix between a carbonate and a silicate endmember (Fig. 3b). Based on two sediment samples with the highest carbonate content and zero

sodium (Table S3), we estimated the magnesium content of carbonates in the area to $\left[\frac{Mg}{Ca}\right]_{br_{carb}} = 0.038 \pm 0.006$ and

$\langle\frac{Mg}{Ca}\rangle_{br_{carb}} = 0.023 \pm 0.004$ where subscript $br$ denotes data from bedrock samples and the angle brackets denote the concentration in weight percent rather than in moles. This low ratio is consistent with the absence of reported dolomite in the area (Chen et al., 2007). Based on knowledge of the carbonate endmember, $\left[\frac{Mg}{Ca}\right]_{br_{carb}}$, and the carbonate content of the samples $\langle Ca_{0.96}, Mg_{0.04}CO_3\rangle$, the $\left[\frac{Na^+}{Ca^{2+}}\right]_{sil}$ and $\left[\frac{Mg^{2+}}{Ca^{2+}}\right]_{sil}$ ratios of the flysch can be expressed as:

$$\left[\frac{Na^+}{Ca^{2+}}\right]_{sil} = \frac{\langle Na\rangle_{br}}{\langle Ca\rangle_{br}-\langle Ca\rangle_{br_{carb}}}\frac{u_{Ca}}{u_{Na}} \tag{10}$$

and

$$\left[\frac{Mg^+}{Ca^{2+}}\right]_{sil} = \frac{\langle Mg\rangle_{br}-\langle Ca\rangle_{br_{carb}}\langle\frac{Mg}{Ca}\rangle_{br_{carb}}}{\langle Ca\rangle_{br}-\langle Ca\rangle_{br_{carb}}}\frac{u_{Ca}}{u_{Mg}} \tag{11}$$

where

$$\langle Ca\rangle_{br_{carb}} = \frac{\langle Ca_{0.96}, Mg_{0.04}CO_3\rangle_{br}}{\frac{u_{CaCO_3}}{u_{Ca}}+\frac{u_{MgCO_3}}{u_{Mg}}\langle\frac{Mg}{Ca}\rangle_{br_{carb}}} \tag{12}$$

And $u_X$ is the molar mass of component $X$ (Bufe et al., 2021). We estimated both ratios based on the assumption that the carbonate content of the bedrock flysch sample with the lowest carbonate content (farthest away from the origin along the regression in $\left[\frac{Na^+}{Ca^{2+}}\right]$-$\left[\frac{Mg^{2+}}{Ca^{2+}}\right]$-space) has a carbonate content of $\langle Ca_{0.96}, Mg_{0.04}CO_3\rangle = 4\text{-}10$ wt%. The minimum value (4%) is from the lowest observed carbonate content in the field (Chen et al., 2007), whereas the upper bound is chosen so that the range of resulting $\left[\frac{Na^+}{Ca^{2+}}\right]_{sil}$ values reaches the global endmember value of 2.7. We then used an average of the resulting ratios estimates with an uncertainty based on half of the range (Fig. 3a-b).

For the cyclic endmember, we used a volume weighted average of rainwater compositions from the eastern flank of Gongga Shan (Jiang et al., 2018) (Table 1, Table S4). The hotspring endmember is also based on hotspring compositions around Mt Gongga (Jiang et al., 2018) (Table S5). All hotsprings are very distinct from rock samples, and notably characterized by $\left[\frac{Na^+}{Ca^{2+}}\right]$-ratios that are $3 - 150$-times higher than the highest measured stream water samples (Fig. 3a, Table S5). Nevertheless, hotspring compositions are highly variable (Fig. 3a), despite a relatively small area that was sampled. In the inversion, we therefore did not sample the hotspring endmembers from a normal distribution around a mean value. Instead, we picked $[Na^+]_{hs}$, $[Mg^{2+}]_{hs}$, and $[Ca^{2+}]_{hs}$ from uniform distributions that span the interquartile range of all hotspring samples. Because of a strong correlation between $[Cl^-]_{hs}$ and $[Na^+]_{hs}$ in these hotsprings, we then picked $[Cl^-]_{hs}$ from the regression between $[Cl^-]_{hs}$ and $[Na^+]_{hs}$.

As an alternative to the inverse model, we decomposed the data with a common forward approach (Bufe et al., 2021; Galy and France-Lanord, 1999; Jacobson and Blum, 2003; Meybeck, 1987; Moon et al., 2014). We assumed that all chloride in the river water samples was atmospherically derived and use precipitation-averaged $\left[\frac{X}{Cl^-}\right]_{cy}$ -ratios of the measured rainwater (Jiang et al., 2018) (Table S4) to adjust the sample concentrations for rain input. We assumed that our rainwater correction accounts for possible anthropogenic input of $SO_4$ from the industrialized Sichuan Basin through acid rain, and we neglected hotspring

contributions. Then, the contributions of silicate and carbonate to the total dissolved $Ca^{2+}$ and $Mg^{2+}$ concentrations were estimated as:

$$[Ca^{2+}]_{sil,fw} = [Na^+]_{spl} \left[\frac{Ca^{2+}}{Na^+}\right]_{sil} \tag{13}$$

$$[Mg^{2+}]_{sil,fw} = [Na^+]_{spl} \left[\frac{Mg^{2+}}{Na^+}\right]_{sil} \tag{14}$$

$$[Ca^{2+}]_{carb,fw} = [Ca^{2+}]_{spl} - [Ca^{2+}]_{sil} \tag{15}$$

$$[Mg^{2+}]_{carb,fw} = [Mg^{2+}]_{spl} - [Mg^{2+}]_{sil} \tag{16}.$$

Equivalent to the inverse model (equations 10 – 11), the silicate endmember ratios for the flysch samples were found by correcting for the carbonate content of the flysch bedrock samples:

$$\left[\frac{Ca^{2+}}{Na^+}\right]_{sil} = \frac{\langle Ca \rangle_{br} - \langle Ca \rangle_{br_{carb}}}{\langle Na \rangle_{br}} \frac{u_{Na}}{u_{Ca}} \tag{17}$$

and

$$\left[\frac{Mg^{2+}}{Na^+}\right]_{sil} = \frac{\langle Mg \rangle_{br} - \langle Ca \rangle_{br_{carb}} \langle \frac{Mg}{Ca} \rangle_{br_{carb}}}{\langle Na \rangle_{br}} \frac{u_{Na}}{u_{Mg}} \tag{18}.$$

### 3.3 Cation sums and fractions

The total sums of cations from silicate and carbonate weathering are:

$$C_{sil} = [Ca^{2+}]_{sil} + [Mg^{2+}]_{sil} + [Na^+] + [K^+] \tag{19}$$

$$C_{carb} = [Ca^{2+}]_{carb} + [Mg^{2+}]_{carb} \tag{20}$$

and in charge equivalents:

$$C_{sil}^{eq} = 2[Ca^{2+}]_{sil} + 2[Mg^{2+}]_{sil} + [Na^+] + [K^+] \tag{21}$$

$$C_{carb}^{eq} = 2[Ca^{2+}]_{carb} + 2[Mg^{2+}]_{carb} \tag{22}$$

The fraction of carbonate weathering is:

$$F_{carb} = \frac{C_{carb}^{eq}}{C_{carb}^{eq} + C_{sil}^{eq}} \qquad (23)$$

The fraction of weathering by sulfuric acid (as opposed to carbonic acid) was estimated from sulfate concentrations (Bufe et al., 2021; Emberson et al., 2018; Galy and France-Lanord, 1999; Torres et al., 2016):

$$F_{sulf} = \frac{2[SO_4^{2-}]_w}{2[SO_4^{2-}]_w + [HCO_3^-]} \qquad (24)$$

where $[SO_4^{2-}]_w$ is the sulfate concentration that is sourced from sulfide oxidation given by:

$$[SO_4^{2-}]_w = [SO_4^{2-}]_{spl} - [SO_4^{2-}]_{cy} - [SO_4^{2-}]_{hs} = [SO_4^{2-}]_{spl} - \alpha_{Ca,cy}\left[\frac{SO_4^{2-}}{Ca^{2+}}\right]_{cy} + \alpha_{Ca,hs}\left[\frac{SO_4^{2-}}{Ca^{2+}}\right]_{hs} \qquad (25)$$

Note that this estimate of $[SO_4^{2-}]_w$ assumes that contributions of sulfate from evaporites are negligible in the area, and that any (likely minor) anthropogenic input as well as minor salts associated with the metasedimentary rocks are captured by the cyclic endmember measured in rainwater samples.

### 3.4 Estimate of denudation rate

Denudation rates were obtained from existing cosmogenic nuclide concentrations of *in situ*-produced [10]Be (Cook et al., 2018; Ouimet et al., 2009). For two sampled catchments (S18.08 and S18.29) without previous denudation rate estimates, we processed two additional samples for analysis of *in situ* [10]Be following the procedure in Cook et al. (2018). Denudation rates were calculated using CRONUS scripts and the Lal/Stone scaling scheme with a sea-level low latitude nucleogenic production rate of 3.7 (Balco et al., 2008) (Table S6). These denudation rates integrate over the time denuding material resides in the first

few upper meters of Earth surface. Although this integration time varies between catchments with different denudation rates, it reflects the timescale of mineral supply to the weathering zone, and is therefore appropriate for comparison to weathering proxies. Where landslides dominate, their stochastic occurrence may disrupt the relationship between soil formation and chemical weathering (Emberson et al., 2016a; Emberson et al., 2016b) and may bias the denudation rate estimates (Chen et al., 2020; Niemi et al., 2005; Tofelde et al., 2018; Yanites et al., 2009). We assume that any such bias is within uncertainty of

the denudation rate measurement. Cosmogenic-derived denudation rates incorporate both physical denudation and chemical weathering, but physical denudation rates commonly greatly dominate denudation in active orogens uplifting siliceous rocks with minor carbonates, such as the ones described here (Dixon and von Blanckenburg, 2012; Erlanger et al., 2021; Gaillardet et al., 1999; Riebe et al., 2001; West et al., 2005). Thus, we assume that increasing denudation rates correspond to increasing physical erosion rates. Finally, in situ [10]Be-derived denudation rates measure the denudation of quartz-containing bedrock.

Based on published bedrock descriptions (Burchfiel et al., 1995; Chen et al., 2007; Jiang et al., 2018; Weller et al., 2013) and observations by the authors (Cook et al., 2018), we assume that carbonate and silicate phases in granitoids and metasediments are mixed to a degree that quartz-derived denudation rates approximate the denudation of the entire rock mass. We acknowledge that unquantified uncertainties may arise where significant weathering occurs below the top few meters of the

regolith (Riebe and Granger, 2013), or where significant differences in quartz mineral contribution within the bedrock occur, but these uncertainties are unlikely to change the major trends observed in this work.

## 4 Results

In the studied catchments, the concentration of dissolved cations (TDC) generally increases with denudation rates (Fig. 4a). However, the strength of that relationship is greatly influenced by lithology (Fig. 4a). Whereas TDC increases 5 – 10-fold in the Triassic Songpan Ganze flysch, the increase of TDC with denudation in the granitic catchments is less than 2-fold and associated with substantial uncertainty (Spearman's p value of 0.19) (Fig. 4a). Mixed flysch – granitic catchments fall in-between the two trends (Fig. 4a). Catchments with Paleozoic metamorphic lithologies have scattered TDC values, many of which fall within the range of values of the Triassic rocks (Fig. 4a). Magnesium and calcium dominate the solute load and constitute 42 – 96% (median 84%) of the TDC. The pattern of increasing TDC is dominated by an increase in calcium concentrations (Fig. 4b), whereas correlations between denudation rate and magnesium, sodium, and potassium concentrations are uncertain (Spearman's p values between 0.1 – 0.9) (Figs. 4c-e). Similarly, no relationship between chloride concentrations and denudation rates is evident (Fig. 4e), but sulfate concentrations show a strong increase with denudation rates in both granites and flysch lithologies (Fig. 4g).

The unmixed cation contributions follow the expectations from the raw data: Cation concentrations from carbonate weathering, $C_{carb}$, and sulfate from sulfide oxidation, $[SO_4^{2-}]_w$, display a strong relationship with denudation (Fig. 5a, c). In the flysch and in the Paleozoic metamorphics they increase strongly with denudation rate, whereas the increase in the granitic catchments is weaker, and the mixed catchments are in-between (Fig. 5a, c). In contrast, cations from silicate weathering, precipitation sources, and hotsprings generally show no relationship to denudation rate for any of the lithologies (Fig. 5b, d-e).

The contribution of carbonate weathering to the total solute load from weathering, $F_{carb}$, strongly increases with denudation irrespective of rock type (Fig. 6a). Even in granitic catchments carbonate weathering contributes 30 – 89% (median 65%) of the cation concentration (Fig. 6a). Thus, both granites and (meta)sedimentary lithologies have comparable fractions of carbonate weathering (Fig. 6a), even if the absolute concentrations differ (Fig. 4). The fraction of weathering by sulfuric acid, $F_{sulf}$, also increases with increasing denudation rate, although there is a large amount of scatter at higher denudation rates (Fig. 6b).

In 24 out of 35 samples (67% of samples), the inferred silicate and/or carbonate cation contributions differ by more than 20% between forward and inverse approaches (Fig. A3). For example, forward-modelled carbonate cation concentrations are more than 20% higher than results from the inversion in 11 samples (31% of samples) and more than 50% higher in 3 samples (9% of samples) (Fig. A3). Silicate cation concentrations inferred from the forward approach are lower in these samples. These samples lie outside of the zone formed between the silicate, carbonate, hotspring, and precipitation endmembers (Fig. 3b) which may explain the offset in the results. Conversely, 9 samples (26%) have more than 20% higher silicate cation

315 concentrations inferred by the forward model (Fig. A3). In these samples, the inversion predicts 3 – 47% (median 13%) cation contributions from hotsprings whereas the median of the cation contributions from hotsprings across all samples is 2%; The hotspring contribution is ignored in the forward approach, and all of the $Na^+$ is attributed to silicate instead, thereby leading to higher silicate cation contributions in the forward model. Despite these differences and resulting outliers, the first order trends described in this contribution are unaffected by the choice of unmixing method (Figs. A4, A5).

320 Our concentration measurements cannot be converted into weathering fluxes, because we lack discharge or runoff data for the sampled catchments. However, in comparison to the nearly three-orders-of-magnitude-wide denudation gradient, the mean annual precipitation does not vary widely across the catchments: 82% of the catchments differ by less than a factor of two and 90% by less than a factor of three of precipitation (Table S1). Further, there is no co-variation between precipitation and denudation rate (Fig. 2b, Table S1). The first-order patterns described above do not change substantially when we consider 325 "inferred" weathering fluxes that are obtained by using mean annual precipitation values as a proxy for runoff (Fig. A6). Therefore, it is unlikely that differences in runoff between catchments strongly affect our data, and we interpret the observed patterns as reflecting the response of the weathering system to changes in denudation fluxes.

## 5 Discussion

Our analysis indicates that increasing denudation rates lead to an increase in carbonate weathering and sulfide oxidation in all 330 sampled lithologies (Figs. 5-6), likely due to the increased supply of carbonate and sulfide minerals during erosion (Bufe et al., 2021; Calmels et al., 2007; Torres et al., 2016). Conversely, silicate weathering appears to be insensitive to denudation rate and represent a small proportion of the total weathering budget (Figs. 5-6). While the granitic catchments have systematically lower cation concentrations from carbonate weathering than the metasedimentary catchments (Fig. 5a), the fraction of carbonate weathering ($F_{carb}$) is increasing with denudation and is of similar magnitude in all rock-types (Fig. 6a). Even in 335 catchments underlain by granitoid rocks, weathering of carbonate phases dominates the solute load at denudation rates above 0.1 mm/yr (Fig. 6a). This finding likely arises from the dissolution of trace carbonate minerals disseminated in the crustal granitoids and metapelites that has been described in a number of active mountain ranges (Blum et al., 1998; Bufe et al., 2021; Emberson et al., 2017; Jacobson et al., 2003; Torres et al., 2016; White et al., 1999). In turn, the insensitivity of silicate cation concentrations to denudation (Fig. 5b) is consistent with a kinetic limitation of silicate weathering (Gabet and Mudd, 2009; 340 West, 2012; West et al., 2005). Such kinetic limitation of silicate weathering in conjunction with increasing sulfide oxidation and carbonate weathering across multiple orders of magnitudes of denudation rates (Fig. 5) has recently been documented in the low-grade metasedimentary rocks of southern Taiwan (Bufe et al., 2021). The data presented here suggest that this pattern applies to metasediments in another mountain range, as well as to granitoid rocks. A series of studies on soil formation in metasedimentary and granitic rocks across denudation rates from $10^{-3}$ mm/y to 2 mm/y has suggested a link between soil 345 denudation rates and soil production rates, and this sensitivity of soil formation rates to denudation has been used to argue for a relationship between orogenesis and silicate weathering (Dixon et al., 2012; Dixon and von Blanckenburg, 2012; Larsen et

al., 2014; Riebe et al., 2004; Riebe et al., 2001). Spanning a comparable range of denudation rates, our data confirm that these findings from soil profiles may not directly translate to the catchment scale (Dixon and von Blanckenburg, 2012; West et al., 2005). This disconnect is due, most likely, to the dilution of soil waters by fluids that drain parts of the landscape that are not soil-covered or follow deep pathways through bedrock. For example, mass movements, such as landslides, that dominate erosion in parts of the study area are expected to alter weathering fluxes (Emberson et al., 2016a). Thus, in active mountain ranges, landscape-scale silicate weathering may be largely insensitive to denudation rates independently of the major lithologies (Bufe et al., 2021; Gabet and Mudd, 2009; West, 2012).

Even though yearly average precipitation rates do not vary considerably across the denudation rate gradient, monthly precipitation can vary by a factor of ~5 – 6 between the high- and low-flow seasons (Jiang et al., 2018). Nevertheless, in small catchments around Mount Gongga, the total dissolved cation concentrations of 80% of rivers is diluted by less than two-fold in the high-flow season (median 1.3-fold) and only $Cl^-$ and $Mg^{2+}$ show substantial dilution (Jiang et al., 2018). Thus, the patterns of concentrations that we describe here should broadly mirror the differences in annual weathering fluxes between catchments. Solute concentrations during the high-flow season are expected to carry an elevated proportion of carbonate weathering with respect to the low flow season (Kemeny et al., 2021; Tipper et al., 2006). Thus, by sampling during the low flow season, and by not accounting for cation exchange with suspended sediment from the clay minerals (Tipper et al., 2021), we might overestimate the proportion of silicate weathering (Kemeny et al., 2021; Tipper et al., 2006; Tipper et al., 2021). The dominance of carbonate weathering over silicate weathering in all catchments (Fig. 6) would, therefore, most likely be strengthened with more data across all seasons and/or with suspended sediment data.

The sulfuric-acid-derived sulfate concentrations $[SO_4^{2-}]_w$ and the cation concentrations from silicate weathering allow to estimate the concentration of $CO_2$ that is sequestered or emitted during chemical weathering (Emberson et al., 2018; Torres et al., 2016). Beyond the calcium carbonate compensation time of ~10 ky; (Zeebe and Westbroek, 2003), the moles of $CO_2$ produced (positive $[CO_2]$) or sequestered (negative $[CO_2]$) during chemical weathering per unit volume of weathering fluid can be expressed as:

$$[CO_2] = [SO_4^{2-}]_w - 0.5 \, C_{sil}^{eq} \tag{23}$$

(Bufe et al., 2021; Torres et al., 2016) (Table S7). Note that this formulation is independent of the proportion of sulfuric acid that weathers carbonates or silicates (see supplement of Torres et al., 2016). The increase of $F_{carb}$ and $F_{sulf}$ with denudation rate (Fig. 6) leads to a clear change from $CO_2$ sequestration to dominantly $CO_2$ release (Fig. 7). These observed trends are most likely explained by the combination of a supply limitation on coupled pyrite oxidation and carbonate weathering and a limitation of silicate weathering rates by the slow dissolution kinetics of silicate minerals (Bufe et al., 2021; Calmels et al., 2007; Gabet and Mudd, 2009; Torres et al., 2016). Eleven catchments fall into the long-term $CO_2$ release field including all but two of the metamorphic catchments (Paleozoic and Triassic), two of the granitoid catchments, and three mixed granitic/metamorphic catchments (Fig. 7). All of these catchments have denudation rates above 0.19 mm/yr. Thus, increasing

denudation rates, lead to decreases in the proportion of weathering that contributes to $CO_2$ drawdown, as sulfate concentrations increase, while silicate cation concentrations are invariable (Figs. 4-6).

Compared to the other lithologies, water samples from granite catchments have substantially lower solute concentrations than waters from (meta)sedimentary catchments (Fig. 4a) and weathering in granitoid catchments remains closest to a balance between $CO_2$ release and $CO_2$ consumption across the entire denudation rate gradient (Fig. 7). This contrast persists when we consider "inferred" weathering fluxes that are obtained using mean annual precipitation values as a proxy for runoff (Fig. A6d). In the absence of major differences in climate or topography between granitic and metasedimentary catchments (Fig. 2), this contrast in concentrations is most likely due to low proportions of minor carbonate and sulfide phases in the granitoid lithologies (Fig. 5a,c). In this case, variations in lithology may change the flux of $CO_2$ drawdown or release by a factor of 2 – 10 at low and high denudation rates respectively (Fig. 7, A6d). Hence, even if the relative pattern of silicate, sulfide, and carbonate weathering with denudation are similar across all lithologies, the absolute weathering fluxes are not. As a consequence, our data suggest that changes in the exposed lithologies across an orogenic cycle can substantially alter weathering fluxes (up to a factor of 2 – 10) independently of variations in denudation rates or runoff.

**6 Conclusion**

Water chemistry data from catchments on the eastern margin of the Tibetan plateau that span across three orders of magnitude in denudation rate illustrate the role of lithology in modulating the link between denudation rate and chemical weathering. Perhaps surprisingly, our data show a uniformity in the first-order relationships between weathering and denudation between different lithologies. In particular, in all lithologies, silicate cation concentrations do not increase with increasing denudation rates whereas concentrations of sulfate and of cations from carbonate do show an increase. Whereas granitic and (meta)sedimentary rocks are characterized by similar fractions of carbonate weathering that are increasing with denudation rate, the (meta)sediments have higher absolute concentrations of sulfate and of cations from carbonate weathering, likely due to higher concentrations of these minor phases in the bedrock. In combination, the weathering reactions lead to a transition from $CO_2$ drawdown to dominantly $CO_2$ release at denudation rates higher than 0.2 mm/yr, that may. In turn, absolute weathering fluxes can vary by a factor of 2 – 10 between catchments draining granites and (meta)sediments with implications for the role of changes in the relative exposure of igneous and sedimentary rocks during mountain growth.

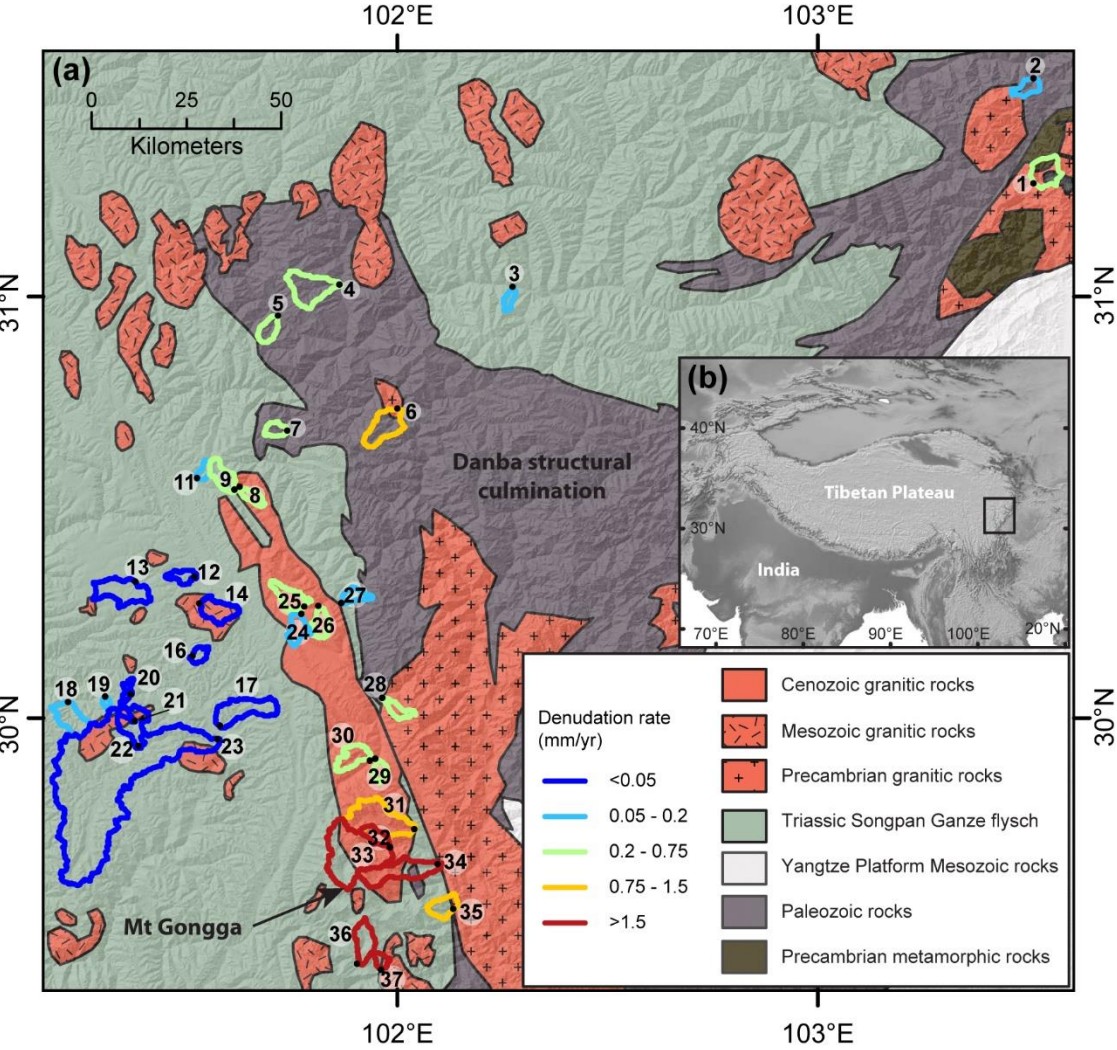

**Figure 1: Overview of the study area**

(a) Simplified geologic map of the study area showing sample locations, sample numbers, and catchment outlines. Outline color indicates
410   [10]Be basin-wide denudation rate from Cook et al. (2018), with two samples from this study (Table S6). (b) Location of study area along the eastern margin of the Tibetan Plateau.

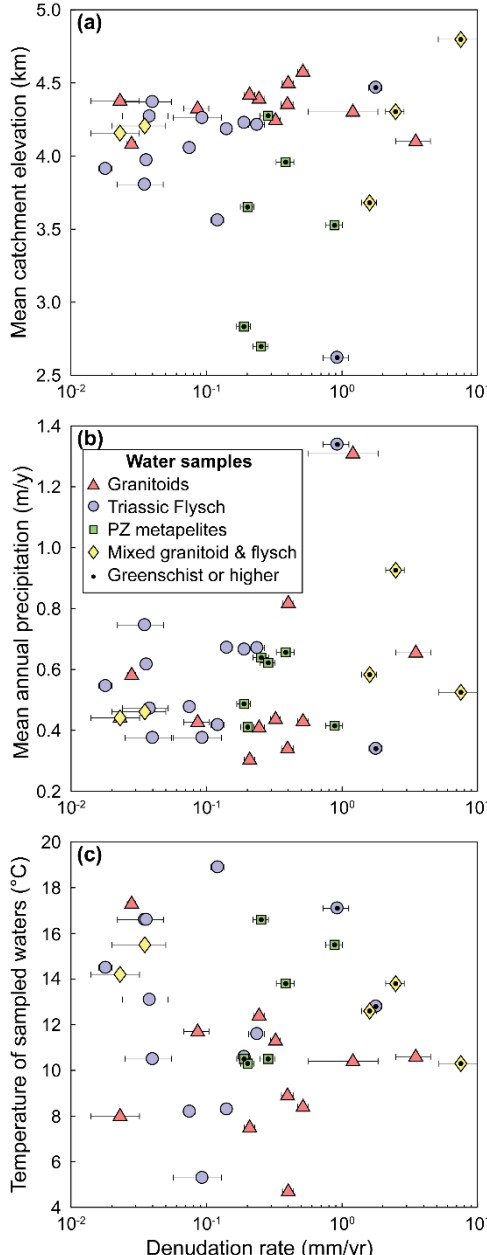

**Figure 2: Variability of topographic and climatic parameter across the denudation gradient**

(a) Catchment-averaged elevation upstream of each sample as a function of denudation rate. (b) TRMM-derived mean annual precipitation in the catchment as a function of denudation rate. (c) Temperature of sampled waters as a function of denudation rate. Catchments underlain by different lithologies are distinguished by symbols and color. Datapoints with a black dot are metamorphosed to greenschist facies or higher.

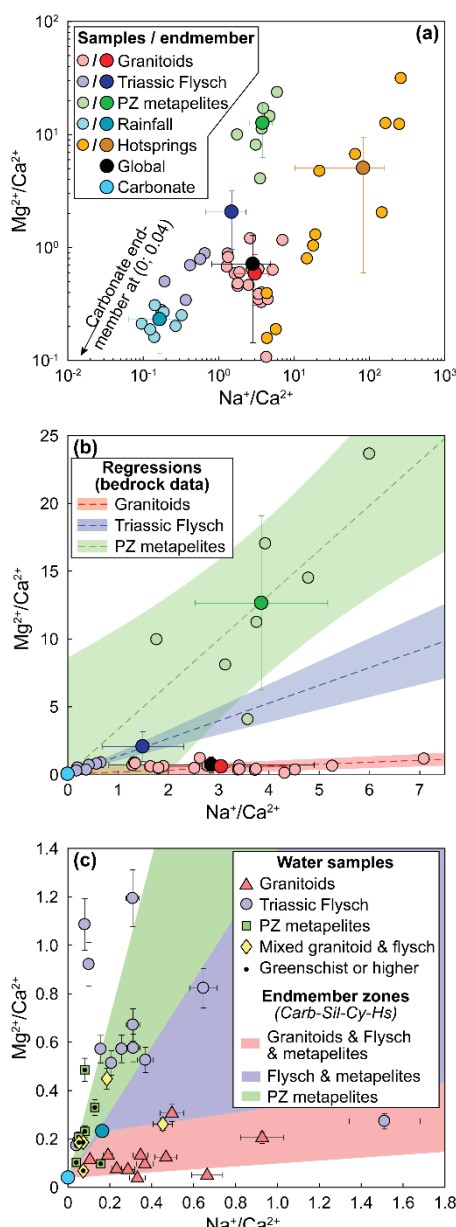

**Figure 3: Endmembers and mixing lines in $Na^+/Ca^{2+}$ – $Mg^{2+}/Ca^{2+}$ space**

(a) Small points mark individual bedrock, rain water, and hotspring chemistry (Table S3) (Chen et al., 2007; Jiang et al., 2018; Weller et al., 2013). Large points with error bars mark corresponding endmember estimates (Table 1). Silicate endmembers for granitoids and Paleozoic metamorphics as well as for the rainfall endmember are estimated from a mean and standard deviation of individual datapoints. The flysch silicate endmember was derived from a regression through the bedrock data and estimates of the carbonate content in the bedrock flysch samples (see text). For hotsprings, the endmember is the median and interquartile range. The carbonate endmember cannot be shown in logarithmic space. (b) Bedrock chemistry and endmembers with regressions through the bedrock data. Colored circle without error bars are lithology data. Colored points with error bars mark the estimate for the silicate endmember for each lithology. Light blue point marks

carbonate endmember. (c) Individual water samples within the endmember space. Colored areas mark the space between the carbonate, silicate, precipitation, and hotspring endmembers for each lithology. Note that spaces are overlapping. Catchments underlain by different lithologies are distinguished by symbols and color. Datapoints with a black dot are metamorphosed to greenschist facies or higher.

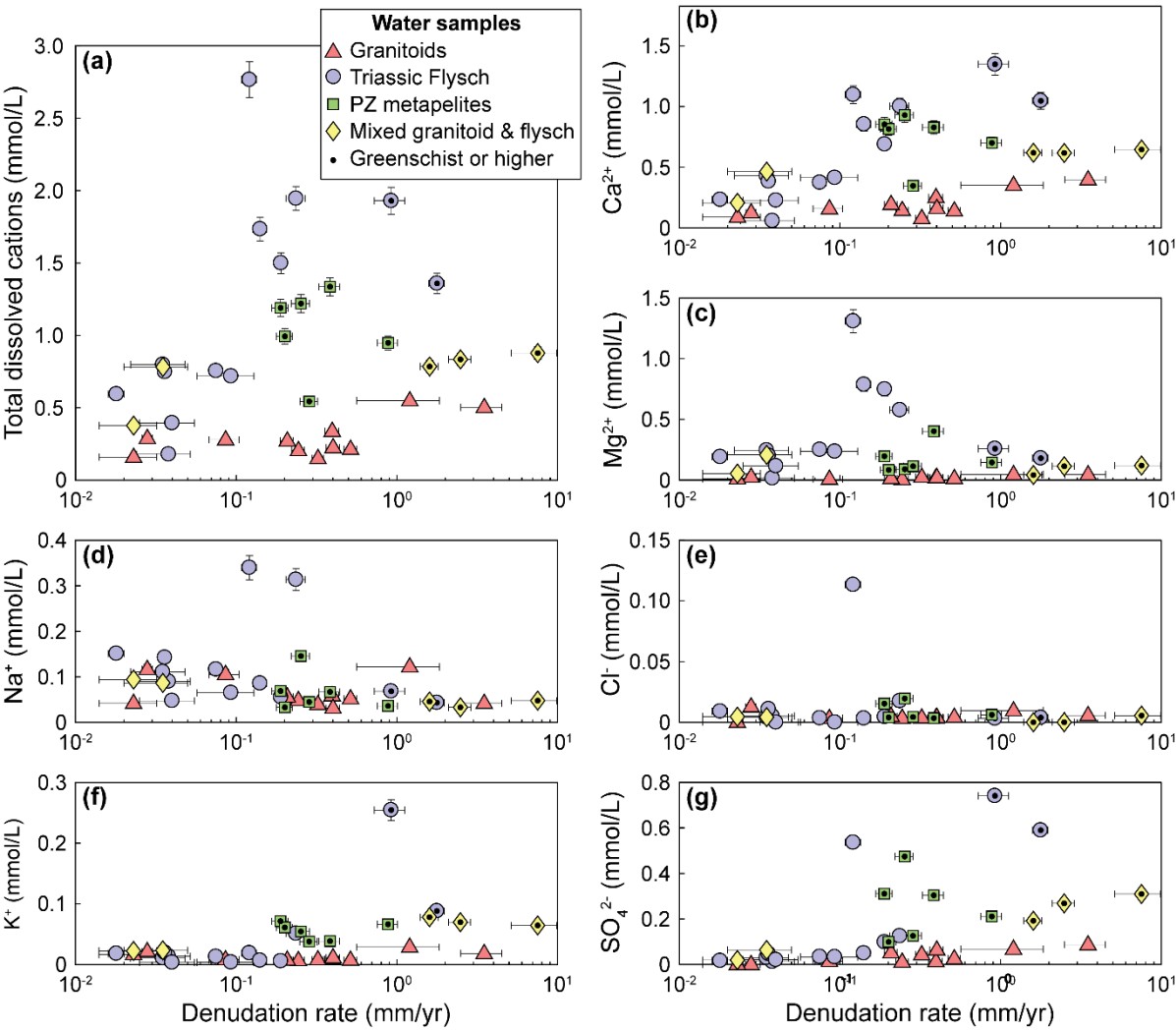

**Figure 4: Solute concentrations vs. denudation rate**

(a) Total dissolved cations. Spearman's rank correlation coefficient and associated p-value: $rho_{granite}^{TDC} = 0.43$, $p_{granite}^{TDC} = 0.19$; $rho_{flysch}^{TDC} = 0.63$, $p_{flysch}^{TDC} = 0.03$. (b) $Ca^{2+}$ concentrations $rho_{granite}^{Ca^{2+}} = 0.63$, $p_{granite}^{Ca^{2+}} = 0.04$; $rho_{flysch}^{Ca^{2+}} = 0.73$, $p_{flysch}^{Ca^{2+}} = 0.006$. (c) $Mg^{2+}$ concentrations. (d) $Na^+$ concentrations. (e) $K^+$ concentrations. (f) $Cl^-$ concentrations. (g) $SO_4^{2-}$ concentrations $rho_{granite}^{SO_4^{2-}} = 0.80$, $p_{granite}^{SO_4^{2-}} = 0.003$; $rho_{flysch}^{SO_4^{2-}} = 0.76$, $p_{flysch}^{SO_4^{2-}} = 0.004$. Catchments underlain by different lithologies are distinguished by symbols and color. Datapoints with a black dot are metamorphosed to greenschist facies or higher.

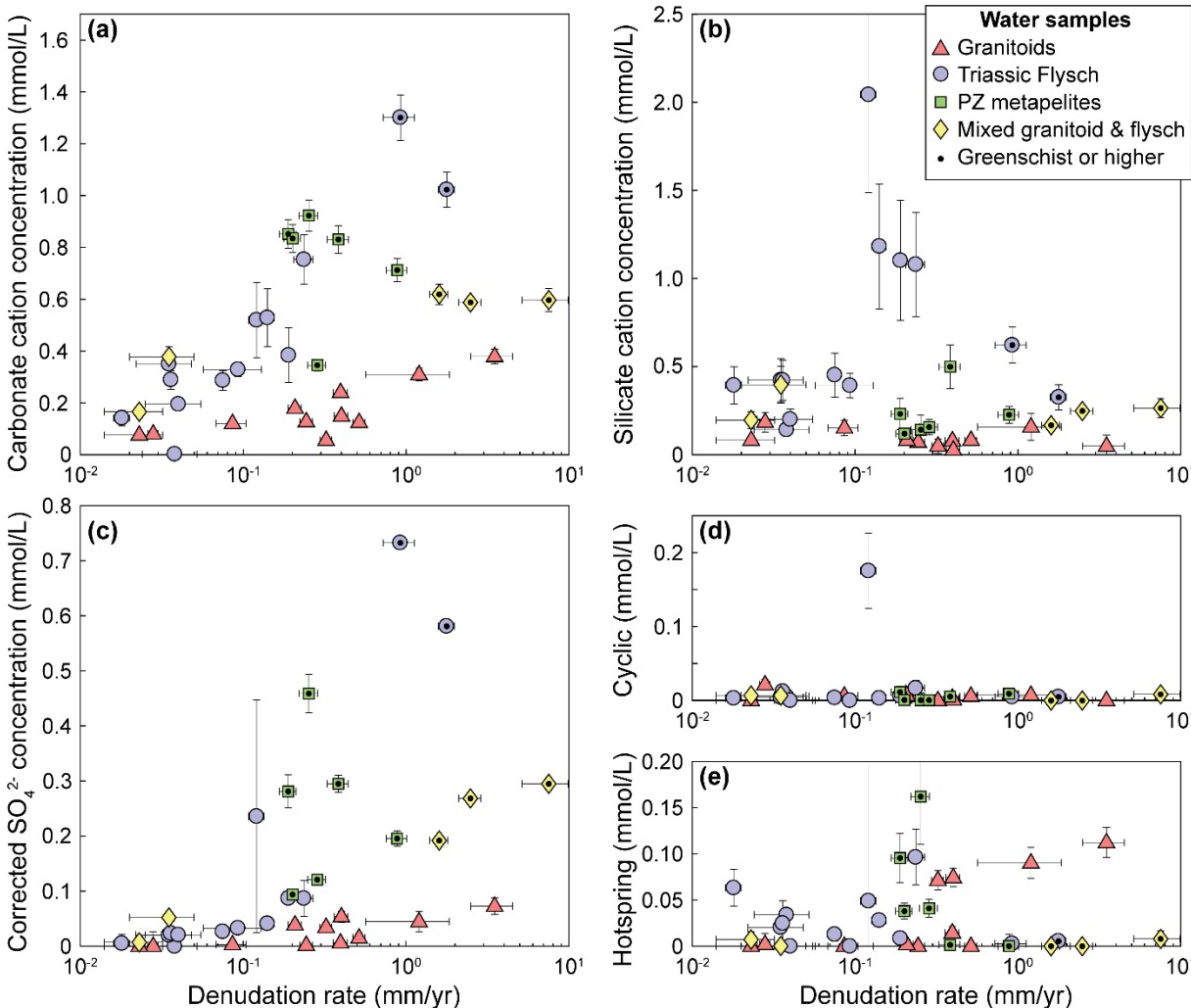

440

**Figure 5: Unmixed contributions to the solute load**

(a) Cation concentrations from carbonate weathering. (b) Cation concentrations from silicate weathering. (c) Sulfate concentrations corrected for precipitation and inferred to derive from sulfide oxidation. (d) Cation concentrations from cyclic sources. (e) Cation concentrations from hotspring sources. Catchments underlain by different lithologies are distinguished by symbols and color. Datapoints

445    with a black dot are metamorphosed to greenschist facies or higher.

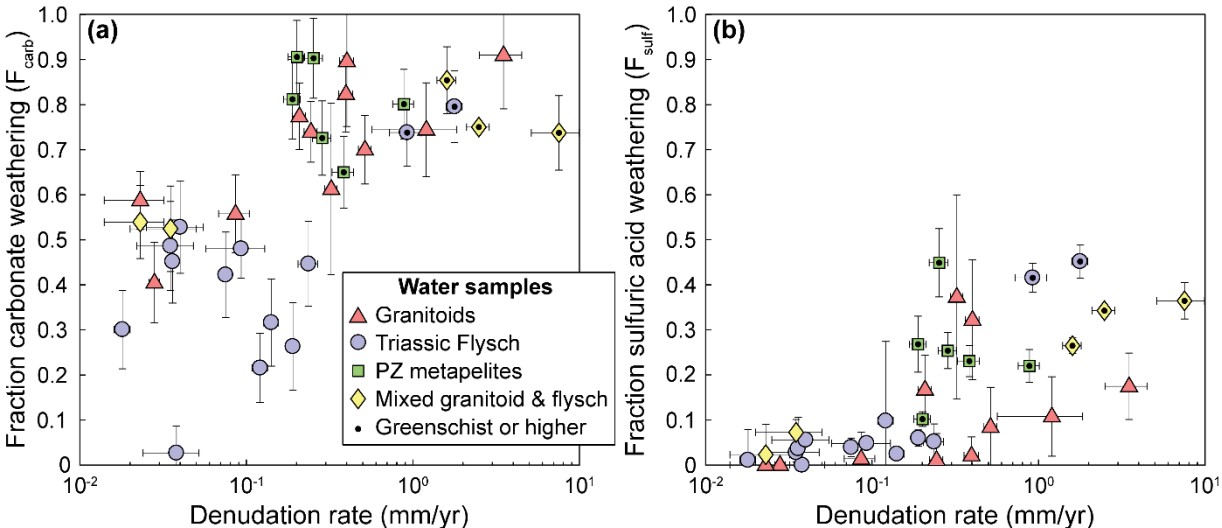

**Figure 6: Fraction of carbonate weathering and weathering by sulfuric acid**

(a) Fraction of carbonate weathering (eq. 23). (b) Fraction of weathering by sulfuric acid (eq. 24). Symbols indicate catchment lithology. Catchments underlain by different lithologies are distinguished by symbols and color. Datapoints with a black dot are metamorphosed to greenschist facies or higher.

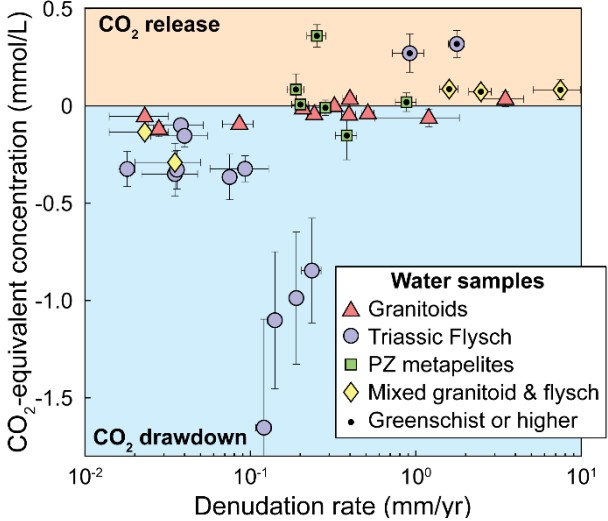

**Figure 7: Influence of weathering on the carbon cycle**

Equivalent concentration of $CO_2$ released or consumed. Symbols indicate catchment lithology. Catchments underlain by different lithologies are distinguished by symbols and color. Datapoints with a black dot are metamorphosed to greenschist facies or higher.

Table 1. Input endmembers for mixing model

| Endmember | Na/Ca | ±Na/Ca | Mg/Ca | ±Mg/Ca | Cl/Ca | ±Cl/Ca | Ca/Na | ±Ca/Na | Mg/Na | ±Mg/Na |
|---|---|---|---|---|---|---|---|---|---|---|
| Silicate: Granite | 3.05 | 1.53 | 0.59 | 0.28 | 0 | 0 | 0.41 | 0.20 | 0.25 | 0.19 |
| Silicate: Paleozoic metapelites | 3.85 | 1.32 | 12.65 | 6.42 | 0 | 0 | 0.29 | 0.13 | 3.38 | 1.43 |
| Silicate: Flysch | 1.49 | 0.81 | 2.06 | 1.10 | 0 | 0 | 0.72 | 0.34 | 1.31 | 0.01 |
| Silicate: Global | 2.86 | 2.04 | 0.71 | 0.57 | 0 | 0 | 0.35 | 0.25 | 0.25 | 0.20 |
| Carbonate | 0 | 0 | 0.038 | 0.006 | 0 | 0 | - | - | - | - |
| Cyclic | 0.17 | 0.1 | 0.23 | 0.12 | 0.98 | 0.41 | - | - | - | - |
| Hotspring | Na/Ca Q1* 10 | Na/Ca Q3** 156 | Mg/Ca Q1* 0.6 | Mg/Ca Q3** 9.5 | Cl/Ca Q1* 2.6 | Cl/Ca Q3** 50.6 | - | - | - | - |

*First quartile

**Second quartile

**Appendix A: Supporting figures**

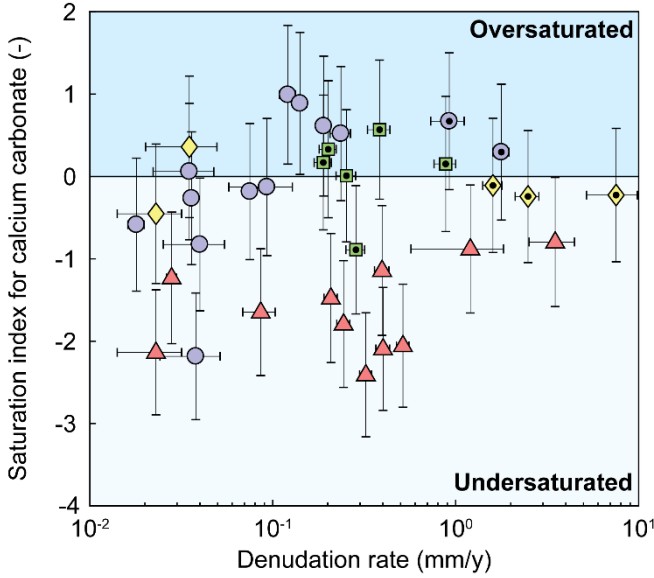

**Figure A1: Saturation index of samples with respect to calcium carbonate**

Catchments underlain by different lithologies are distinguished by symbols and color. Datapoints with a black dot are metamorphosed to
465 greenschist facies or higher.

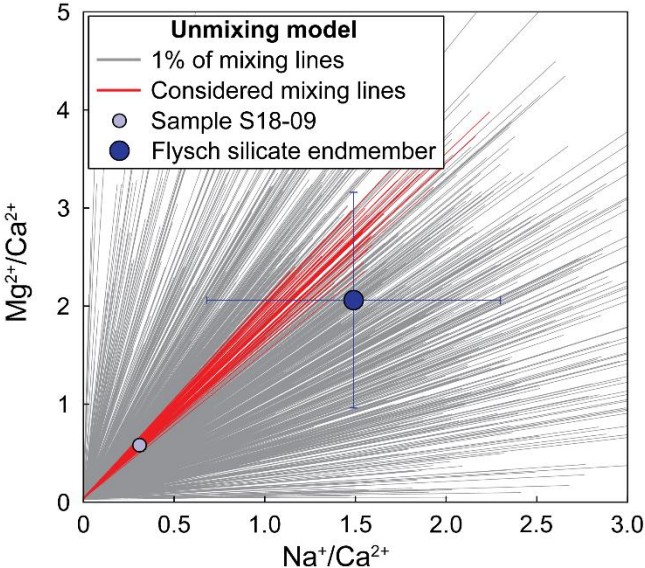

**Figure A2: Example of mixing lines from Monte Carlo modelling in Na⁺/Ca²⁺ – Mg²⁺/Ca²⁺ space**

**Figure A2: Example of mixing lines from Monte Carlo modelling in $Na^+/Ca^{2+}$ – $Mg^{2+}/Ca^{2+}$ space**

Monte Carlo Iterations of the mixing model for sample S18-09 (small light-blue point). Flysch silicate endmember and 1σ uncertainty in
dark blue. 1000 out of 100,000 (1%) of mixing lines from the Monte Carlo modelling marked as grey lines. Red lines show all runs with
470 $\chi^2_{total} \leq 1$ among those 1% of runs.

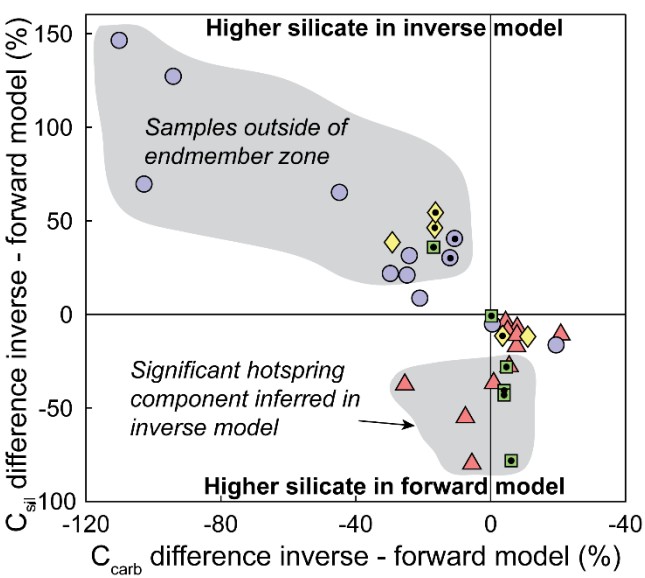

**Figure A3: Difference between forward and inverse unmixing approaches**

Positive values indicate that the inverse approach predicts higher cation ratios than the forward approach. All samples in the upper left quadrant are outside of the carbonate – silicate – hotspring – precipitation endmember space in $Na^+/Ca^{2+}$ – $Mg^{2+}/Ca^{2+}$ space (Fig. 3c). Samples with lower inverse silicate cation concentrations are characterized by 3-47% hotspring contribution that is not accounted for in the forward model (see text). Catchments underlain by different lithologies are distinguished by symbols and color. Datapoints with a black dot are metamorphosed to greenschist facies or higher.

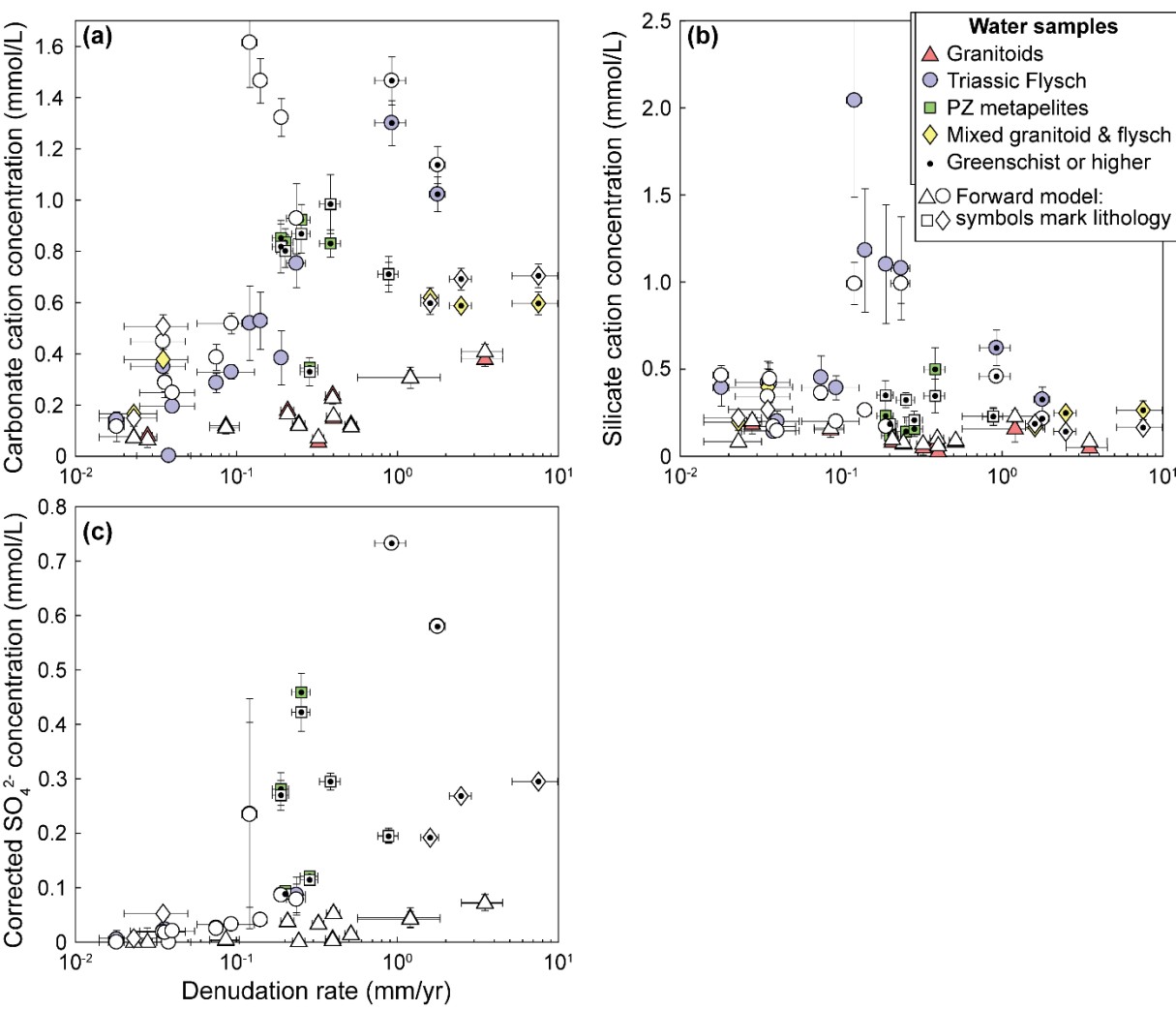

**Figure A4: Forward model results for silicate, carbonate, and sulfide weathering contributions**

Data from Fig. 5 plotted with results from the forward model (white points). Lithology in the data from the forward model is distinguished by symbology only. Where results from forward and inverse approaches are similar, symbols overlap.

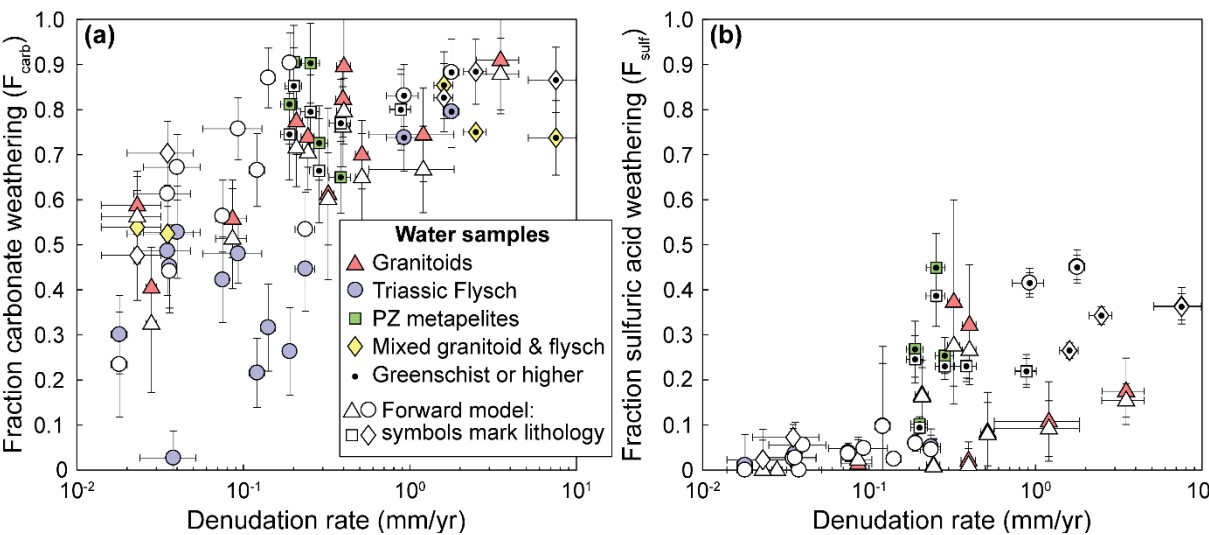

485

**Figure A5: Forward model results for the fraction of carbonate weathering and weathering by sulfuric acid**

Data from Fig. 6 plotted with results from the forward model (white points). Lithology in the data from the forward model is distinguished by symbology only. Where results from forward and inverse approaches are similar, symbols overlap.

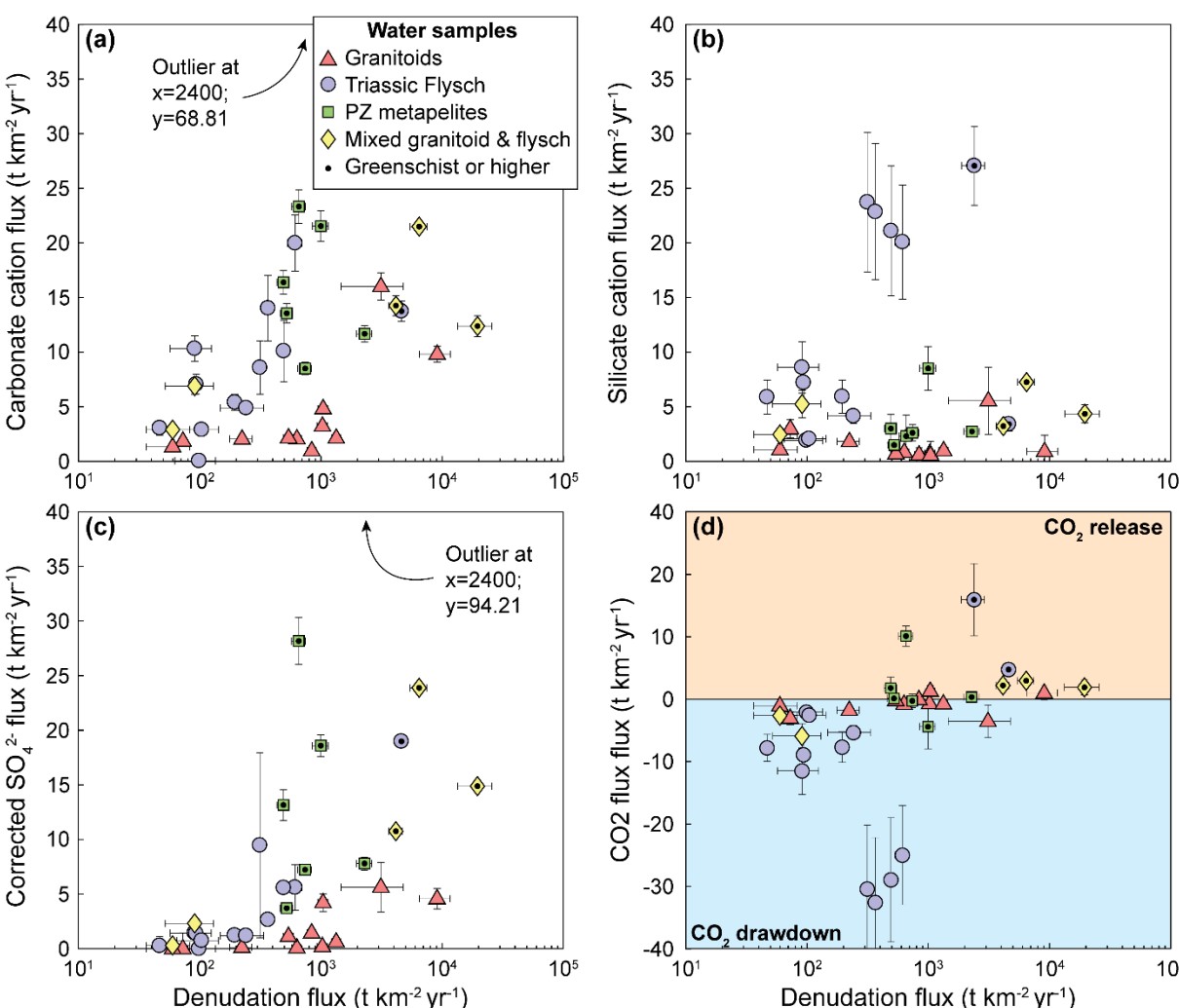

490

**Figure A6: Inferred weathering fluxes**

(a-c) Data from Fig. 5a-c and (d) data from Fig. 7 converted to fluxes by using mean annual precipitation as runoff estimate. One outlier with exceptional precipitation values is not plotted on panels a and b.

## Data Availability

All data used in this paper are included as tables in the Supplement.

## Author contribution

K.L.C. and N.H. conceived the study and collected samples in the field. A.B., H.W., and A.G. contributed to laboratory analyses. K.L.C. and A.B. led the data analysis and interpretation and wrote the manuscript with input from all authors.

## Competing interest

The authors declare that they have no conflict of interest.

## Acknowledgements

We thank Fan Xuanmei, Dai Lanxin, and Chen Jie for help with fieldwork logistics. Bernhard Zimmermann, Jutta Schlegel, Daniel Frick, Andrea Gottsche, Tanya Goldberg, and the HELGES Lab are thanked for laboratory assistance. We thank J. Jotautas Baronas and an anonymous reviewer for detailed and constructive comments on an earlier version of the manuscript.

## Financial support

AB has received funding from the European Union's Horizon 2020 research and innovation programme under the Marie Sklodowska-Curie grant agreement No 841663. Publication was supported within the funding programme "Open Access Publikationskosten" Deutsche Forschungsgemeinschaft (DFG, German Research Foundation) - Project Number 491075472.

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
