# Peer review of "The effect of lithology on the relationship between denudation rate and chemical weathering pathways. Evidence from the eastern Tibetan Plateau."

_Earth Surface Dynamics, 2021_

## Author Comment (AC1)

**Comments by the editor**

This manuscript has now received two detailed reviews, one anonymous, and one by J. Baronas (very detailed). Both these reviews provide a positive evaluation of the manuscript. Overall, I agree with he reviewers that this is a well written manuscript and well constructed piece of science and I encourage you to submit a response to the reviewers and a revised manuscript.

*Thank you very much for facilitating the review of the manuscript. We now have gone through all of the reviewers' comments and replied to them point-by-point. Note that the comments are formulated in a way that reflects our already implemented changes in the manuscript.*

*In addition to the changes explained in the point-by-point replies, we re-ran the inversion with a slight change in endmember values that is due to taking the mean rather than the median of the rock endmember values as well as correcting a mistake in the silicate-endmember estimate for the flysch samples. This change modified the detailed position of the datapoints but did not alter the overall patterns described in the manuscript.*

Although the comments by the reviewers are relatively straightforward to address, they are important and will help get this manuscript the visibility that it deserves.

All the comments of the reviewers be considered as a point by point basis and fully addressed in the revised manuscript. Based on my reading of the manuscript, and the reviewers, I suggest the following points be given careful consideration.

Missing references are highlighted by both reviewers. I think their suggestions are valid, and if possible, should be incorporated.

*We incorporated most suggested references as explained in our responses.*

R2 suggests that a more specific title might be useful, potentially providing a greater impact for this work and I think this is worth considering very seriously.

*We modified the title to read: "The effect of lithology on the relationship between denudation rate and chemical weathering pathways. Evidence from the eastern Tibetan Plateau."*

The source of sulfate raised by R2 is important. As a minimum I suggest the uncertainty associated with the assumption that all SO4 is from pyrite be taken into account. What about hot springs for example?

*Following the comment by R2, we removed the note on the potential differences between metamorphic and non-metamorphic sediments as explained in the detailed responses.*

The points raised by R1 about the inclusion of rain and hot springs in the inversion model but then were corrected for confused me also. This needs clarification.

*We modified and expanded the methods which hopefully clarifies all of the concerns and confusions that were caused by this section.*

---

## Author Comment (AC2)

**Reviewer #2 J. Jotautas Baronas**

General comments

This manuscript by Bufe et al. presents new river chemistry data from a range of small catchments in the eastern Tibetan plateau, using it to partition silicate-carbonate-sulfide weathering and the relative effect on atmospheric CO2. Using previously published cosmogenic denudation data, they demonstrate that carbonate-sulfide weathering correlates with denudation, whereas silicate weathering does not. The results are consistent with a growing body of evidence that, put bluntly, mountains around the world may be acting as CO2 sources rather than sinks. This is an important and contentious question, and therefore any new constraints are welcome. In this sense, it is a topic within the scope of ESurf, with substantial new data and conclusions worthy of publication.

From technical standpoint, the manuscript is very well written and structured, with good use of English language. The title needs to be more specific, however. The previous/related work is properly accredited/cited with just a few exceptions as noted below. From a scientific standpoint, the study provides a "snapshot" of pre-monsoon weathering stoichiometry in the study area, allowing interrogation of relative spatial trends. A weakness (as would be in any river chemistry study) is the lack of runoff or seasonal timeseries data. Regardless, it is a little-studied and uninstrumented area, the data are therefore valuable, and they quite convincingly support the discussion and the conclusions (although I provide some suggestions below where I think the authors can better recognize and state these limitations). The authors partly compensate for this with a comparison of both inverse and forward modelling approaches, which is not done very frequently in such studies and is useful. In summary, it is a welcome contribution, which I recommend for publication, once the issues raised below have been addressed.

*Thank you very much for carefully reading through the manuscript and providing critical but constructive comments that helped to more clearly address the limitations of the work. We hope that the implemented changes successfully address your concerns.*

Major issue 1: sources of sulfate

Because this is, in a sense, an "old school" approach that relies solely on solute concentrations (as opposed to isotopic signatures of sulfate and/or bicarbonate), the authors need to be as convincing as possible that riverine sulfate is indeed (very likely) derived exclusively from sulfide weathering.

- On L126 authors cite previous papers to say "major evaporites are not reported". But because evaporite phases have dissolution kinetics even faster than carbonates or sulfides, even trace amounts could end up dominating the riverine budgets. For example, how confident are the authors that there is no carbonate-associated-sulfate (CAS) in the local rocks, perhaps similarly to elsewhere in the Himalaya (https://doi.org/10.1016/j.sedgeo.2021.106027, recognizing the differences in the lithologies and tectonic histories)? Presumably, the extent of petrologic/mineralogic studies in Sichuan is considerably lower? All this is to say, that "no sulfate in rocks" appears like a major assumption, which needs to be backed as thoroughly as possible, given all available data. If a convincing enough argument cannot be made, then perhaps the modelling needs to incorporate an appropriate degree of uncertainty in this regard.

*The lack of sulfur isotope data is certainly a limit to the study. That said, a contribution of evaporite salts to the sulfate from weathering of the sedimentary rocks is likely minor:*

- *Except for one sample, chloride concentrations in the samples range between 4-20 μmol/L and are thus well within the range of concentrations expected for atmospheric chloride input of <30 μmol/L [Gaillardet et al., 1999]. Moreover, the ratio of $\left[\frac{Ca^{2+}}{Na^+}\right]$ concentrations are higher than expected for typical evaporite deposits [Gaillardet et al., 1999].*
- *Chen et al. [2007] analyzed compositions of clastic metasediments around the Danba structural culmination (including the study area) and include both elemental analyses and optical investigation. They do not report evaporite rocks or salt components in the analyzed rock samples.*
- *Jiang et al. [2018] sampled waters in the same area and argued that the contribution of salts is likely minor, because Cl and SO4 are not correlated. We also do not find any relationship between Cl and SO4 in our data.*

*We did not mention carbonate associated sulfates (CAS) (i.e. sulfate that replaces a carbonate in the calcite lattice), because it is a very minor component of carbonates. Modern carbonates have CAS of ~ 0.1 – 1wt%. In the rock record CAS constitutes 0 – 0.1 wt% of the carbonate [Burdett et al., 1989; Gill et al., 2008]. In all of our samples, the ratio of $SO_4\_w/C\_Carb$ is >1 wt% with values of 10 – 80% sulfate for most samples. Thus, the contribution of CAS should not be more than 10% for the samples with the lowest $SO_4$ and is most likely much less than 1% for most of the samples – and therefore within the uncertainty of our $SO_4\_w$ estimate.*

*We modified the section as follows: We assume that evaporites are a minor component of the dissolved solids: (i) Existing petrologic and geochemical studies do not report evaporite deposits in the upper Triassic flysch or in the Paleozoic metamorphic rocks around the Danba Structural culmination [Chen et al., 2007; Jiang et al., 2018], (ii) except for one sample, chloride concentrations range between 4-20 μmol/L and are thus well within the range of concentrations expected for atmospheric chloride input [Gaillardet et al., 1999], and (iii) the ratio of $\left[\frac{Ca^{2+}}{Na^+}\right]$ concentrations are higher than expected for typical evaporite deposits [Gaillardet et al., 1999; Chen et al., 2007; Jiang et al., 2018].*

- This all relates to another point that the authors make quite prominently. L24: "sulfur oxidation state during prograde metamorphism of pelites in the mid crust could lead to sulfate reduction that is even more complete than in low-grade sediments and provides a larger sulfide source"; L346: "The production of sulfide by thermal reduction of trace marine sulfates embedded in marine sediments becomes efficient at temperatures above ~500 °C (Goldstein and Aizenshtat, 1994), and could be a viable explanation." Doesn't this imply that in the less metamorphosed flysch units there should be unreduced marine sulfate? Perhaps there is a good explanation but currently this seems like a major contradiction with the "no sulfate" assumption.

*We agree that this point was poorly supported and confusing. The idea behind it was that sulfate-rich fluids could have enriched the metamorphic rocks during metamorphism. This idea still raises the question where those fluids come from. In addition, the difference between metamorphic and non-metamorphic erosion-sulfate concentration trends is small at best and indistinguishable at worst because both regressions are within the confidence bands*

*(Fig. 7A in the previous manuscript version). Given the lack of a clear difference in the data and the lack of a satisfying explanation for a source of additional sulfate, we decided to remove this part of the argument and focus on the strong signals contained in the data (the trends in concentrations of carbonate, silicate and sulfide weathering and the differences and similarities between granitoid and metasediments)*

Major issue 2: discussion of concentrations & geomorphic controls on weathering

- You mention on L97 that precipitation (and therefore probably runoff) rates vary only by a factor 4-5 across the catchments and you therefore disregard them given the much higher variability in denudation. But crucially, your solute concentrations for the most part vary within a factor of 4-5 or so, and therefore could theoretically be strongly modified by differences in runoff among the catchments. Granted, you are careful to avoid any discussion of weathering "fluxes" in your study. But even if you do not explicitly say it, solute concentrations do effectively stand in for fluxes (or rather area-normalized rates) in parts of the discussion. Strictly speaking, only solute ratios (and the model results) are immune to this, whereas, for example, an assertion that absolute rates (fluxes) of pyrite and carbonate weathering increase with D, could be seen as unsupported. However, I do not think that any extrapolation to rates or fluxes (implied or explicit) is unwarranted and it is an important part of the discussion (see below). If we can't discuss solute concentrations, then only Fcarb and Fsulf are left, which is not that much… My suggestions would be to 1) acknowledge a bit better the caveat of the lack of runoff data (and concentration-runoff timeseries, for that matter); and 2) plot the mean annual precipitation rates somewhere to better convince the reader that differences in runoff are not driving the observed differences in concentrations.

*The lack of runoff data is the other key limitation to the study. In order to address this limit, we did indeed plot the precipitation rates (Fig. 2b). To further strengthen the argument, we now include a version of the key data plots in the appendix in which all of the concentrations are multiplied by the precipitation (as the best available proxy for runoff differences). The major patterns described in this manuscript appear insensitive to variations in runoff.*

*We now add at the end of the results section: "In comparison to the nearly three-orders-of-magnitude-wide denudation gradient, the mean annual precipitation does not vary widely across the catchments: 82% of the catchments differ by less than a factor of two and 90% by less than a factor of three in precipitation (Table S1). Further, there is no co-variation between precipitation and denudation rate (Fig. 2B, Table S1). Finally, the observed first-order patterns described above do not change substantially when we estimate weathering fluxes by using mean annual precipitation values as a proxy for runoff (Fig. A5). Therefore, it is unlikely that differences in runoff between catchments strongly affect our data, and we interpret the observed patterns as reflecting the response of the weathering system to changes in denudation fluxes."*

- L302-309 raises and briefly discusses the second major point of this study – that silicate weathering rates appear decoupled from denudation in the study area. I feel it deserves quite a bit more attention (keeping in mind the caveats raised in the previous point). As you note, your findings contradict some of the soil production literature which has observed both a positive and a negative (Ferrier & Kirchner, 2008)

relationship between erosion and weathering. Where do your study catchments fall on these global relationships of W vs D (recognizing the large caveats in estimating W here)? What could be the mechanisms decoupling soil-scale from catchment-scale weathering rates?

*A full exploration of this decoupling between soil and water chemistry is beyond the scope of this study. That said, we tried to clarify the note on the potential mechanism behind the disconnect and write: "This disconnect is due, most likely, to the dilution of soil waters by fluids that drain parts of the landscape that are not soil-covered or follow deep pathways through bedrock. For example, mass movements such as landslides that dominate erosion in parts of the study area are expected to alter weathering fluxes [Emberson et al., 2016].*

*As you note, without runoff data, we cannot get the weathering fluxes. We now note that the range of denudation rates is comparable between the soil studies and our study. Further, we estimate silicate weathering fluxes using precipitation rates as a runoff estimate (New Fig. A5).*

- To enable this more thorough discussion (and more generally), at least a brief (even if anecdotal) description of the geomorphology of the sampled catchments would be very useful. Are there any constraints on the soil / regolith depths? Is erosion landslide-dominated? Are these catchments vegetated or above the tree line? If nothing else, there are DEM data for steepness etc, already probably discussed in Cook et al 2018.

*We added information on relief and slopes as well as position of the tree line to the manuscript.*

Other points
- The title is arguably too broad for the scope of the paper and should include the study location, eg. "… in eastern Tibetan Plateau"

*Was changed*

- In study site description, please mention which major river basins the studied headwater catchments belong to.

*Added*

- Missing literature
  - Spence & Telmer 2005, GCA (https://linkinghub.elsevier.com/retrieve/pii/S0016703705005946) – one of the seminal "modern" papers outlining the implications of coupled carbonte-sulfide weathering to atmospheric CO2, as well as the erosional control on it
  - Kemeny et al 2021, GCA (https://doi.org/10.1016/j.gca.2020.11.009) – a thorough study of sulfide-carbonate-silicate weathering across the Nepal Himalayas, with some insights and implications applicable in the context of this manuscript. For example, independently constraining Fsulf for silicates and carbonates, demonstrating that it is higher for carbonates (not explicitly discussed but see their Fig. 7e). They also demonstrate strong seasonal

variations in overall Fsulf – theoretically a similar dynamic could be expected in Songpan-Ganze.

- o Relph et al 2021, EPSL (https://doi.org/10.1016/j.epsl.2021.116957) - demonstrating both erosional and lithological control of pyrite weathering rates in the Mekong basin (headwaters draining southeastern Tibetan plateau, not a world away from the study area in this manuscript), see Section 5.5, Fig. S12

*Added*

- Eq 19 – first of all, I believe there is a typo and the subscript of the last parameter should say "sil" as opposed to "carb" (otherwise, there needs to be a full derivation of this equation and an explanation how silicate weathering does not affect atmospheric CO2! ðŸ˜ Š ). Secondly, I think it is important to discuss an assumption behind this formulation, namely that Fsulf is equal for both carbonates and silicates. Is this a reasonable assumption? It seems to me there are a few factors suggesting that perhaps not: 1) carbonates and silicates are often lithologically associated, as you mention in your manuscript and as documented elsewhere (eg CAS); 2) carbonate dissolution kinetics are much faster than silicate kinetics; 3) Kemeny et al. 2021 demonstrating Fsulf being higher for carb compared to sil using an inversion with more degrees of freedom. Therefore, perhaps valuable to explore what happens if this assumption is relaxed, allowing sulfide-carbonate to first consume one of the reactants, only then switching to either sulfide-silicate or carbonic-carbonate weathering (depending on the excess reactant). This would provide an upper limit on potential CO2 release.

*Yes, there was a typo. Thank you very much for catching that.*

*This (perhaps slightly simplified) formulation does not assume any partitioning of sulfuric acid between silicates and carbonates. Essentially the formulation assumes that, to first order, all that matters is that sulfuric acid consumes alkalinity where it doesn't matter where that alkalinity comes from. The supplement of Torres et al. [2016] explains that framework in detail, and the equation 19 is from that supplement. We added an explicit reference to that supplement.*

Technical
- L41 – for posterity, should add "over timescales longer than marine carbonate compensation" or similar.

*Added*

- L136: typo, X = Mg2+ repeated twice, should probably be Cl-

*Yes – thanks!*

- L135-150 needs to be worded more clearly. The Monte Carlo component comes out a bit of nowhere. Do I understand correctly that you do the minimization 100,000 times for each set of samples (catchments) with a given silicate lithology? Or separately for each river sample? Should L147 say "[X/Ca]riv"? Do you mean that the squared distances are normalized by variance for each ratio, so that all ratios are weighted equally? What is a "mean input value"? The mean, or "best" a priori estimate of each end-member ratio? Maybe it's best to show the chi-square formula, given there are

two sets of parameters used to calculate it, and it seems this is where the hydrothermal springs are disregarded (as mentioned on L192), if I understand correctly. Also it is unclear how analytical [X/Ca]riv uncertainties are included here. Are some of the modelled values outside of the analytical uncertainty range (which is what by the way)? Are the [X/Ca]riv and [X/Ca]endmember residuals given equal weight in the chi-squared? If so, what's the logic behind it, and why not give the actual measured [X/Ca]riv bigger weight? How many runs pass your criteria in the end, and are used to estimate the overall uncertainty? Are the "best values" selected from a single run with the lowest chi-square, or some average? Please assume that the reader would like to reproduce your procedure exactly, without having to read the previous papers, and write accordingly all the steps (in supplementary/appendix if necessary). I know this technique has been around for decades, but honestly, the actual implementation details are almost always skipped and they always differ slightly. Reproducibility is important.

*This is a fair point and we tried to clarify the section.*

- L172: shouldn't the carbonate stoichiometry be "Ca0.96 Mg0.04", given the preceding sentence? I find it confusing to use a 1:1 calcite/dolomite ratio here and elsewhere.

*Sorry – the notation was confusing. The concentration actually does not assume a stoichiometry – the ratio is taken from* $\left[\frac{Mg^{2+}}{Ca^{2+}}\right]_{carb} = 0.04$. *We changed the notation to* $\langle Ca_{0.96}, Mg_{0.04}CO_3 \rangle$.

- L172: do "square brackets" refer to "⟨ ⟩" -- if so, this is not correct, as square brackets are "[]" already used for the dissolved concentrations.

*Yes – that was a typo and was changed to angle brackets*

- Eq 6-7: where do the "riv" values come from? Do they indicate riverine sediment, or somehow dissolved concentrations (in which case this would be very confusing and apparently circular logic)? If sediment, then I strongly suggest using a different subscript.

*The subscript was meant to denote a sampled ratio – in this case the bedrock sample which was obviously very confusing. We now changed all of the subscripts denoting water sample to* $spl$ *and all of the bedrock sample to* $br$.

- Eq 8-11: what does the subscript "spl" refer to?

*We now changed all of the riv subscripts to spl (denoting a water sample)*

- L306: should say "dilution" rather than "dissolution"?

*Yes - thanks*

- L323: "ref 62" – wrong citation format

*Yes - thanks*

- Fig 3a: point sizes seem all the same in the actual plot

*Thanks - fixed*

- Fig 3b: the title says this is a regression through the bedrock data (which sounds like a fit to all the individual bedrock samples), whereas the caption says these are carbonate-silicate mixing lines. Which one is it?

*fixed*

- Fig 3c: gray circle must be the carbonate endmember but this is not noted anywhere. It should also be easier to see (more distinct color/symbol)

*fixed*

- Fig A2: should say "lower silicate weathering" at the bottom?

*Yes - fixed*

**References**

Burdett, J. W., M. A. Arthur, and M. Richardson (1989), A Neogene seawater sulfur isotope age curve from calcareous pelagic microfossils, *Earth Planet. Sc. Lett.*, *94*(3), 189-198, doi:https://doi.org/10.1016/0012-821X(89)90138-6.

Chen, Y., F. Liu, H. Zhang, L. Nie, and L. Jiang (2007), Elemental and Sm-Nd isotopic geochemistry on detrital sedimentary rocks in the Ganzi-Songpan block and Longmen Mountains, *Frontiers of Earth Science in China*, *1*(1), 60.

Emberson, R., N. Hovius, A. Galy, and O. Marc (2016), Chemical weathering in active mountain belts controlled by stochastic bedrock landsliding, *Nat. Geosci.*, *9*(1), 42-45, doi:10.1038/ngeo2600.

Gaillardet, J., B. Dupré, P. Louvat, and C. J. Allègre (1999), Global silicate weathering and CO2 consumption rates deduced from the chemistry of large rivers, *Chem. Geol.*, *159*(1), 3-30, doi:http://dx.doi.org/10.1016/S0009-2541(99)00031-5.

Gill, B. C., T. W. Lyons, and T. D. Frank (2008), Behavior of carbonate-associated sulfate during meteoric diagenesis and implications for the sulfur isotope paleoproxy, *Geochim. Cosmochim. Ac.*, *72*(19), 4699-4711, doi:https://doi.org/10.1016/j.gca.2008.07.001.

Jiang, H., W. Liu, Z. Xu, X. Zhou, Z. Zheng, T. Zhao, L. Zhou, X. Zhang, Y. Xu, and T. Liu (2018), Chemical weathering of small catchments on the Southeastern Tibetan Plateau I: Water sources, solute sources and weathering rates, *Chem. Geol.*, *500*, 159-174, doi:https://doi.org/10.1016/j.chemgeo.2018.09.030.

Torres, M. A., A. J. West, K. E. Clark, G. Paris, J. Bouchez, C. Ponton, S. J. Feakins, V. Galy, and J. F. Adkins (2016), The acid and alkalinity budgets of weathering in the Andes–Amazon system: Insights into the erosional control of global biogeochemical cycles, *Earth Planet. Sc. Lett.*, *450*, 381-391, doi:http://dx.doi.org/10.1016/j.epsl.2016.06.012.

---

## Author Comment (AC3)

**Reviewer #1**

This manuscript explores how the relationship between riverine solute geochemistry and denudation rate varies between catchments with different bedrock compositions. The authors find that the concentrations of silicate-derived cations are relatively constant with increasing denudation whereas contributions from carbonate weathering and sulfide oxidation increase with increasing denudation rate. The authors also find differences in sulfate concentrations between lithologies at a given denudation rate, which affects the calculated amount of CO2 drawdown.

The results of this study are fairly non-controversial. Nevertheless, it is nice to actually observe in data some patterns that may have been predicted/expected. Moreover, many similar datasets lack the tight constraints on denudation from 10Be measurements that are available for this study. While one could nitpick about some of the assumptions that go into the mixing model (i.e., no secondary mineral formation and the congruent dissolution of bulk silicate rock), I think the conclusions that the authors come to are the most parsimonious and that their approach is sufficient to explain the major trends in the data. Accordingly, I think the manuscript is appropriate for publication after some minor revisions. In particular, there were some methodological details that I was confused about that could be explained better in a revised version of this manuscript.

*We want to thank R1 for taking the time to carefully read through and comment on the manuscript*

Line 42: It is probably better to be more precise here and state that carbonate weathering by carbonic acid is CO2 neutral over timescales longer than the characteristic timescale of carbonate precipitation in the ocean.

*Added*

Line 43: I would recommend against the Lasaga 1984 reference here. There are many other options out there that report measurements of silicate mineral dissolution rates. I would also suggest Johnson et al. (2019) as a more recent reference on pyrite oxidation kinetics. Lastly, it might also be worth citing the work by Kanzaki et al. (2020) on the reactive-transport modeling of silicate weathering and pyrite oxidation.

Johnson, Aleisha C., et al. "Experimental determination of pyrite and molybdenite oxidation kinetics at nanomolar oxygen concentrations." Geochimica et Cosmochimica Acta 249 (2019): 160-172

Kanzaki, Yoshiki, Susan L. Brantley, and Lee R. Kump. "A numerical examination of the effect of sulfide dissolution on silicate weathering." Earth and Planetary Science Letters 539 (2020): 116239

*We changed the Lasaga [1984] reference to Berner [1978].*

*Johnson et al. [2019] investigates pyrite oxidation conditions of the Archean – so we believe that Williamson and Rimstidt [1994] is more applicable to the modern setting that we discuss here.*

*The work of Kanzaki et al. [2020] is very interesting, but it goes into details of the coupling between silicate and sulfide weathering that is beyond the scope of this conceptual introduction.*

Line 50: It might make sense to cite Ibarra et al. 2016 here as well as it also compares basaltic and granitic weathering fluxes.

Ibarra, Daniel E., et al. "Differential weathering of basaltic and granitic catchments from concentration–discharge relationships." Geochimica et Cosmochimica Acta 190 (2016): 265-293.

*Added*

Line 144: My interpretation is that, depending upon the geologic map data, one of three different potential silicate end-members was used for each river sample as opposed to, for example, trying all three different silicate end-members for each catchment. I would appreciate a very clear statement about which data constraints were applied to which catchments just to avoid any confusion.

*We revised the description of the inversion which hopefully clarifies the point.*

Line 184: "We corrected all major elements for atmospheric inputs…".  I am confused by this. I thought that rainwater was an end-member in the set of mixing equations described at the start of section 3.2. This sentence here makes it sound like the data were corrected for rainwater contributions and then inverted for carbonate vs. silicate contributions. It would be helpful if the authors could clarify their exact approach.

*Thanks for the comment. Indeed, the phrasing was not very clear. We changed it to say "For the cyclic endmember, we used a volume weighted average of rainwater compositions from the eastern flank of Gongga Shan"*

Line 192: "…we did not consider the hot spring end-member in finding the best-fit model in the inversion". This confused me. The start of section 3.2 describes a four end-member mixing model (silicate, carbonate, rain, and hydrothermal inputs). However, this sentence makes it seem like hydrothermal inputs were completely ignored such that the authors actually use a three end-member mixing model. If that is the case, I think it is very confusing to describe a hydrothermal end-member only to ultimately ignore it. Again, it would be helpful if the authors could clarify exactly how potential hydrothermal contributions were considered.

*This sentence was very unclear. Following your comments and those from R2 we revised the description of the inversion which hopefully clarifies this point.*

Line 284: I am not sure if the authors *have* to make this argument that concentrations are proportional to fluxes. I think it is best to stick to what is actually measured (i.e., concentrations and concentration ratios) as opposed to making untested assumptions about

discharge variations based on imperfect proxies (mean annual rainfall) that do vary quite considerably (factor of 3) given the range of concentration variability.

*A similar point was made by R2 and the lack of runoff data remains one of the primary limits of this study. Assuming that the range of precipitation values reflects – at least to first order – the expected range of runoff, we can estimate the likely importance of differences in runoff between catchments. Approximating weathering fluxes by using the mean annual precipitation does not change the observed pattern of the data with respect to denudation rates substantially (see new Figure A5). It therefore seems likely that runoff differences play a minor role in modulating the observed variability of water chemistry with denudation in the study area. We now write: "Further, there is no co-variation between precipitation and denudation rate (Fig. 2B, Table S1). Finally, the first-order patterns described above do not change substantially when we consider 'approximate weathering fluxes' that are obtained by using mean annual precipitation values as a proxy for runoff (Fig. A5). Therefore, it is unlikely that differences in runoff between catchments strongly affect our data, and we interpret the observed patterns as reflecting the response of the weathering system to changes in denudation fluxes."*

Line 307: "… to the dissolution of soil waters by fluids from other parts of the landscape…". This sentence was confusing to me. I am not sure how fluids dissolve soil waters. I recommend that it be edited for clarity.

*Sorry for the confusion. This is a typo and meant to say "dilution".*

Line 316: It might make sense to cite Kemeny et al. 2021 here as well given that they also looked at seasonal changes in the carbonate weathering fraction at a similar site.

Kemeny, Preston Cosslett, et al. "Sulfate sulfur isotopes and major ion chemistry reveal that pyrite oxidation counteracts CO2 drawdown from silicate weathering in the Langtang-Trisuli-Narayani River system, Nepal Himalaya." Geochimica et Cosmochimica Acta/294 (2021): 43-69

*Added*

**References**

Berner, R. A. (1978), Rate control of mineral dissolution under Earth surface conditions, *Am. J. Sci.*, *278*(9), 1235-1252, doi:10.2475/ajs.278.9.1235.

Johnson, A. C., S. J. Romaniello, C. T. Reinhard, D. D. Gregory, E. Garcia-Robledo, N. P. Revsbech, D. E. Canfield, T. W. Lyons, and A. D. Anbar (2019), Experimental determination of pyrite and molybdenite oxidation kinetics at nanomolar oxygen concentrations, *Geochim. Cosmochim. Ac.*, *249*, 160-172, doi:https://doi.org/10.1016/j.gca.2019.01.022.

Kanzaki, Y., S. L. Brantley, and L. R. Kump (2020), A numerical examination of the effect of sulfide dissolution on silicate weathering, *Earth Planet. Sc. Lett.*, *539*, 116239, doi:https://doi.org/10.1016/j.epsl.2020.116239.

Lasaga, A. C. (1984), Chemical kinetics of water-rock interactions, *J. Geophys. Res.*, *89*(B6), 4009-4025, doi:10.1029/JB089iB06p04009.

Williamson, M. A., and J. D. Rimstidt (1994), The kinetics and electrochemical rate-determining step of aqueous pyrite oxidation, *Geochim. Cosmochim. Ac.*, *58*(24), 5443-5454, doi:http://dx.doi.org/10.1016/0016-7037(94)90241-0.

---

## Author Response (AR2)

**Comments by the editor**

This is a nice manuscript, and I am satisfied that the authors have addressed the comments from the first round of reviews and adapted the manuscript accordingly. Upon re-reading, there was one point that I wasn't clear on in the revised manuscript. On lines 202-203, referring to data excluded based on the chi-squared distances, it states that 91% of the model runs were excluded. Is this really correct? Effectively this means the model is failing to converge in 91% of the Monte-Carlo simulations? I imagine (hope) this is a typo. If not a typo, as a minimum it warrants a clearer explanation of what is going on and potentially a more significant re-think. It won't make a significant difference to the conclusions of the manuscript, but it is important to get right.

*Thank you for going through the manuscript again and pointing out this lack of clarity. The 91% was not a typo. The large number of runs that do not enter in the estimate of the parameters for the mixing model is due to our "brute-force" approach to the modelling without any optimization. With each iteration, we pick groups of endmembers from the entire possible space without consideration of results from previous iterations. Because the endmember space is large, many runs do not fit the data within the chosen threshold of $\chi^2_{total} \leq 1$. Model efficiency and the number of iterations that fit could be increased by implementing an optimization method, such as the one presented by Moon et al. [2014], but that should not alter the results fundamentally.*

*In order to clarify this section, we now write: The uncertainty in these parameters was estimated from all Monte Carlo runs that fit the data within a threshold of $\chi^2_{total} \leq 1$ (on average 9% of runs). The high number of runs above the threshold of $\chi^2_{total} > 1$ is linked to our approach of picking groups of endmembers independently for each Monte Carlo iteration from the entire endmember space (see example in Fig. A2) without an optimization (e.g. Moon et al., 2014).*

*We also include a new figure (Figure A2) that attempts to illustrate this approach.*

**References**

Moon, S., C. P. Chamberlain, and G. E. Hilley (2014), New estimates of silicate weathering rates and their uncertainties in global rivers, *Geochim. Cosmochim. Ac.*, *134*, 257-274, doi:http://dx.doi.org/10.1016/j.gca.2014.02.033.